# Atmospheric pressure mass spectrometric imaging of live hippocampal tissue slices with subcellular spatial resolution

Jae Young Kim[1], Eun Seok Seo[1], Hyunmin Kim [2], Ji-Won Park[3], Dong-Kwon Lim[4] & Dae Won Moon[1]

We report a high spatial resolution mass spectrometry (MS) system that allows us to image live hippocampal tissue slices under open-air atmospheric pressure (AP) and ambient temperature conditions at the subcellular level. The method is based on an efficient desorption process by femtosecond (fs) laser assisted with nanoparticles and a subsequent ionization step by applying nonthermal plasma, termed AP nanoparticle and plasma assisted laser desorption ionization (AP-nanoPALDI) MS method. Combining the AP-nanoPALDI with microscopic sample scanning, MS imaging with spatial resolution of 2.9 μm was obtained. The observed AP-nanoPALDI MS imaging clearly revealed the differences of molecular composition between the apical and basal dendrite regions of a hippocampal tissue. In addition, the AP-nanoPALDI MS imaging showed the decrease of cholesterol in hippocampus by treating with methyl β-cyclodextrin, which exemplifies the potential of AP-nanoPALDI for live tissue imaging for various biomedical applications without any chemical pretreatment and/or labeling process.

[1] Department of New Biology, DGIST, Daegu, 42988, Republic of Korea. [2] Companion Diagnostics and Medical Technology Research Group, DGIST, Daegu, 42988, Republic of Korea. [3] Graduate School of Analytical Science and Technology (GRAST), Chungnam National University, Daejeon, 34134, Republic of Korea. [4] KU-KIST Graduate School of Converging Science and Technology, Korea University, Seoul, 02841, Republic of Korea. Correspondence and requests for materials should be addressed to D.W.M. (email: dwmoon@dgist.ac.kr)

Recently, atmospheric pressure (AP) ionization mass spectrometry, which can directly analyze samples with minimal or no sample preparation, has fascinated researchers in fields other than analytical chemistry because it seems to have potential applications in various research fields[1–10]. Applications have been demonstrated for pesticide or explosive detections and acquisition of molecular information in biological samples using various MS methods at AP[11–13]. In addition, AP mass spectrometry (AP-MS) imaging, which can provide both chemical and spatial information has also been studied with a number of different AP-ionization methods. The desorption electrospray ionization (DESI) method[10,14], AP matrix-assisted laser desorption ionization (AP-MALDI) method[15], and laser ablation electrospray ionization (LAESI)[16] method have been regarded as prominent AP desorption/ionization methods capable of producing MS imaging of biological samples. However, owing to their limited spatial resolution, these AP-MS imaging techniques still face several challenges for practical bioimaging applications[17]. Since the current ionization sources for AP-MS analysis are strongly dependent on the state of ionized gas or spray state of charged droplet, further improvement of the spatial resolution by reducing the size of source device is quite limited. Thus, the spatial resolution of AP-MS imaging in practice remains in the several-tens-to-hundred micrometers range, hindering precise analysis of small and complex biological samples[18–21].

In order for AP-MS to be more widely used for biomedical research, we need to develop an AP-MS subcellular imaging technique that can efficiently ablate the bulk molecular constituents down to tens of μm to obtain unprejudiced information reflecting the whole-tissue composition.

In this paper, we report a high spatial resolution AP-MS imaging method (called AP-nanoPALDI MS) that can provide subcellular imaging of living biological tissue slices with a sampling depth down to several tens of μm. The AP-nanoPALDI MS analysis for live mouse hippocampal tissue slices provided information on the distribution of lipids and metabolites in the apical and the basal dendrite regions without any chemical or fixative treatment. It also showed the decrease of cholesterol in hippocampus by treating with methyl β-cyclodextrin. The AP-nanoPALDI MS imaging technique is expected to be applicable to various label-free bioimaging applications, such as accurate molecular histology for fresh tissues and tissue-based drug screening.

## Results

### AP-nanoPALDI MS system for high-resolution MS imaging.

Shown in Fig. 1a, b is the developed AP-nanoPALDI MS system, which consists of a mass analyzer, a sampling stage, a femtosecond (fs) laser oscillator, an AP plasma equipment, and airflow-assisted ion transport equipment. The whole AP-nanoPALDI MS system for this study is described in detail in Supplementary Fig. 1. To optically monitor the biological sample and decide the analysis region, a stage of an inverted optical microscope (IX73, Olympus, Japan) was used as a sampling stage for desorption and ionization procedures, as shown in Fig. 1a.

The employed laser oscillator without an optical amplifier generates ultrafast laser pulses with a pulse repetition rate of 75 MHz and pulse energy of 2 nJ at 802 nm. Details of the laser system are described in the Methods section. An fs near infrared (NIR) laser beam was introduced into the inverted microscope by installing a dichroic beam splitter (FF720-SDi01, Semrock, USA). It allowed optical imaging monitoring and laser desorption from a biological sample simultaneously without a separate objective lens, as shown in Fig. 1c. All the MS images presented in this study were obtained using a ×20 objective lens (NA = 0.45; LUCPlanFLN 20X, Olympus, Japan), and the diameter of the focused laser beam could be reduced to 1.2 μm (see Supplementary Note 1). For precise positioning of a region of interest (ROI) at the micrometer level and sample moving at a constant velocity for raster scanning, a programmable motorized XY scanning stage was mounted on the inverted optical microscope.

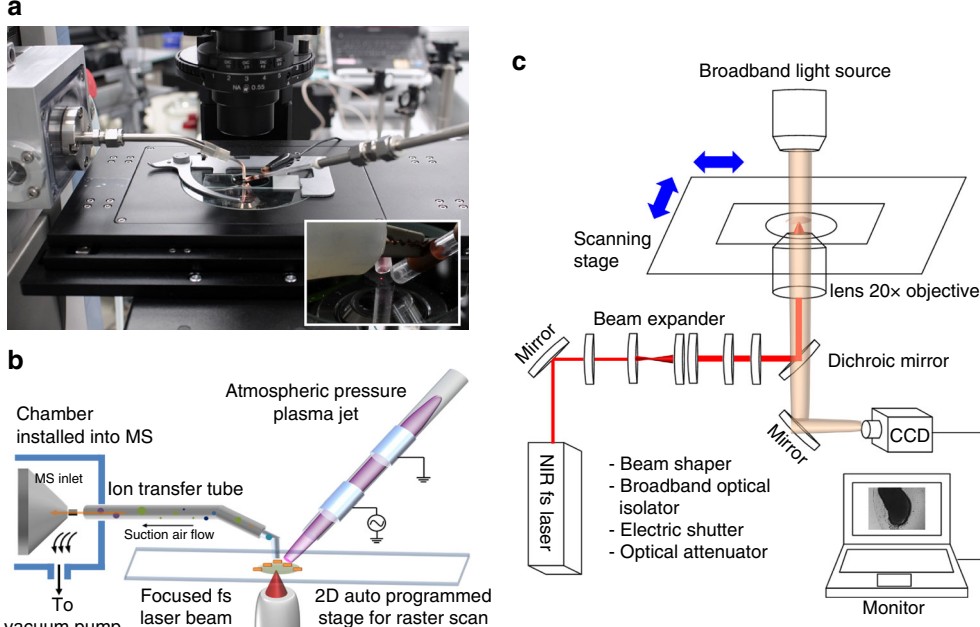

**Fig. 1** AP-nanoPALDI mass spectrometric system with a combination of fs laser oscillator, atmospheric pressure (AP) plasma jet, and airflow-assisted ion transfer equipment. **a** Picture of analysis stage in proposed AP-nanoPALDI MS system. **b** Schematics of AP desorption and ionization procedure on stage and ion transport. **c** Different light paths between visible and near infrared (NIR) spectra using dichroic mirror. This feature allows the NIR laser light to be reflected towards the target while the visible light passed through freely. Therefore it allows optical imaging monitoring and desorption by laser beam of the biological sample simultaneously without additional focusing lens

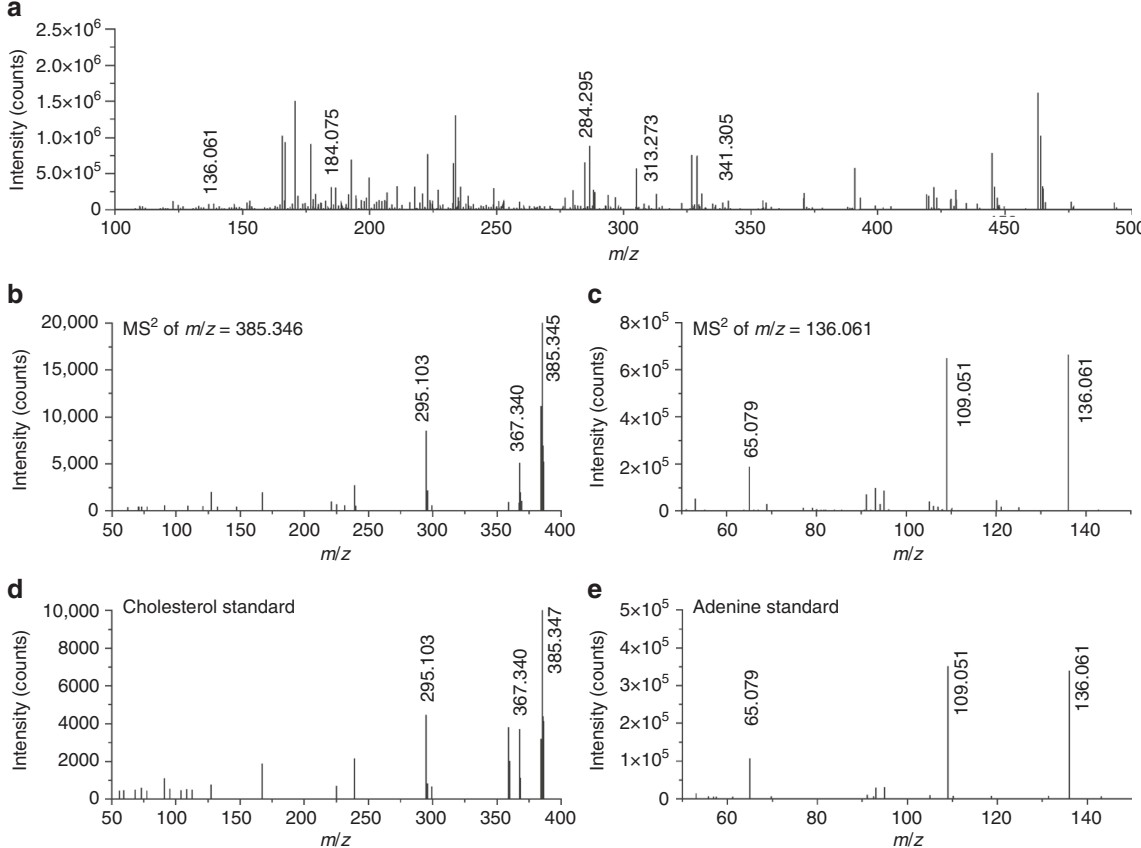

**Fig. 2** AP-nanoPALDI mass spectra from a mouse hippocampal tissue slice and standard materials in the positive ion mode. **a** Mass spectra from a mouse hippocampal tissue slice in the range of $m/z = 100–500$. The plasma background has been subtracted from the spectra. AP-nanoPALDI MS/MS of **b** compound at $m/z = 385.346$ in the hippocampal tissue slice, **c** compound at $m/z = 136.061$ in the hippocampal tissue slice, **d** standard cholesterol, and **e** standard adenine

The nonthermal AP helium plasma jet device, consisting of a tube with discharge gases and electrodes, formed a helium plasma medium above the sample, as described in Fig. 1b. The neutral molecules desorbed by the focused laser compulsorily meet the helium plasma medium by arranging two desorption/ionization sources, and some were ionized by metastable helium atoms with excitation energies of 19.8 eV[22].

As shown in Fig. 1a, b, the additional pumping system, known as airflow-assisted ion transfer equipment[23–25], was installed into the mass analyzer for effective transport of molecules and ions. The airflow-assisted ion transfer equipment consisted of an ion transfer tubing, a chamber, and a dry pump, as shown in Supplementary Fig. 2a, b. The pump generated airflow inside the ion transfer tube toward the MS inlet, as shown in Supplementary Fig. 2c, d. With the airflow-assisted ion transfer equipment, the intensity of ions increased roughly 40 times compared to the no pump condition, as shown in Supplementary Fig. 3.

In addition to the airflow-assisted ion transport, gold nanorods (rod-shaped gold nanoparticles, AuNRs) as characterized in Supplementary Fig. 4, were necessary for an efficient and stable desorption procedure by fs NIR laser oscillator. AuNRs are well known to serve as not only a photon energy reservoir but also a thermal energy transfer material to analytes[26]. While most biological analytes do not effectively absorb NIR light[27], AuNRs absorb NIR light due to their longitudinal resonance given by their aspect ratio and convert the photon energy from laser beam into thermal energy[28]. Even though the tissue slice was treated by AuNRs, uptake of AuNRs is a natural biological behavior mainly caused by endocytosis of live tissues. Thus, tissue slices with

AuNRs remained viable until MS analysis began. The AuNRs in the live tissue slices can interact with fs laser light, resulting in a significantly enhanced desorption procedure and higher mass spectra intensity. Without AuNRs, no mass spectra could be observed.

**Mass spectra from live hippocampal slice of adult mouse.** Adult-mouse hippocampal tissues were chosen for live tissue imaging, as this not only allows a clear distinction among sub-regions, such as cornu ammonis 1 (CA1), cornu ammonis 3 (CA3), and dentate gyrus (DG), structurally, but also plays a very important role in consolidating information of short-term memories to long-term memories and spatial navigation[29–32]. The term "live tissue slices" in reference to "short-term living tissue slices" is well-established in pseudo-two-dimensional models for research into neurophysiology, pathophysiology, and electrophysiology, in particular for studies in stretch-activated ion channels (SAC), microelectrode arrays (MEAs) of various orga-notypic tissues[33–35]. In addition, even after treatment of AuNRs into the live hippocampal tissue slices, they still remained viable until desorption and ionization procedures. The detailed pre-paration of live hippocampal tissue slices and treatment of AuNRs for this study are described in the Methods section.

The ions detected by AP-nanoPALDI method from the live hippocampal tissue slice of an adult Institute of Cancer Research (ICR) mouse are presented in Fig. 2a. Because the plasma is an ionized gas, many ions were detected from the plasma itself; thus, the plasma background has been excluded from spectra. The MS spectra, as shown in Fig. 2a, represented the average of 400

**Table 1 Assigned lipids, metabolites, and derivatives from mouse hippocampal tissue slices in the positive ion mode using AP-nanoPALDI MS**

| Compound | Measured $m/z$ | Theoretical $m/z$ | Error (ppm) | Molecular formula | Species | Ref. |
|---|---|---|---|---|---|---|
| Sterol lipid | | | | | | |
| Cholesterol | 385.3459 | 385.3464 | 1.30 | $C_{27}H_{45}O$ | $[M - H]^+$ | 37 |
| | 369.3521 | 369.3515 | −1.62 | $C_{27}H_{45}$ | $[M + H - H_2O]^+$ | 39 |
| Glycerolipid | | | | | | |
| MAG 16:1 | 311.2575 | 311.2581 | 1.93 | $C_{19}H_{35}O_3$ | $[M + H - H_2O]^+$ | 36 |
| | 329.2692 | 329.2686 | −1.82 | $C_{19}H_{37}O_4$ | $[M + H]^+$ | 71 |
| MAG 16:0 | 313.2731 | 313.2737 | 1.92 | $C_{19}H_{37}O_3$ | $[M + H - H_2O]^+$ | 36 |
| | 331.2851 | 331.2843 | −2.41 | $C_{19}H_{39}O_4$ | $[M + H]^+$ | 71 |
| MAG 18:2 | 337.2732 | 337.2737 | 1.48 | $C_{21}H_{37}O_3$ | $[M + H - H_2O]^+$ | 36 |
| | 355.2851 | 355.2843 | −2.25 | $C_{21}H_{39}O_4$ | $[M + H]^+$ | 71 |
| MAG 18:1 | 339.2889 | 339.2894 | 1.47 | $C_{21}H_{39}O_3$ | $[M + H - H_2O]^+$ | 36 |
| | 357.3007 | 357.2999 | −2.24 | $C_{21}H_{41}O_4$ | $[M + H]^+$ | 71 |
| MAG 18:0 | 341.3046 | 341.3050 | 1.17 | $C_{21}H_{41}O_3$ | $[M + H - H_2O]^+$ | 36 |
| | 359.3161 | 359.3156 | −1.39 | $C_{21}H_{43}O_4$ | $[[M + H]^+$ | 71 |
| Sphingolipid | | | | | | |
| Ceramide 18:0 | 548.5390 | 548.5401 | 2.01 | $C_{36}H_{70}NO_2$ | $[M + H - H_2O]^+$ | 72 |
| Sphingosine | 282.2785 | 282.2791 | 2.13 | $C_{18}H_{36}NO$ | $[M + H - H_2O]^+$ | 73 |
| | 300.2905 | 300.2897 | −2.66 | $C_{18}H_{38}NO_2$ | $[M + H]^+$ | 73 |
| Sphinganine | 284.2945 | 284.2947 | 0.70 | $C_{18}H_{38}NO$ | $[M + H - H_2O]^+$ | 73 |
| | 302.3062 | 302.3054 | −2.65 | $C_{18}H_{40}NO_2$ | $[M + H]^+$ | 73 |
| Adenine | 136.0614 | 136.0617 | 2.20 | $C_5H_6N_5$ | $[M + H]^+$ | 53 |
| Phosphocholine | 184.0754 | 184.0733 | −11.41 | $C_5H_{15}NO_4P$ | $[M]^+$ | 40 |

The table includes exact mass measurements of ion peaks in the mouse hippocampal tissue slice by AP-nanoPALDI-QE-Orbitrap (number of C atoms: degree of unsaturation)

consecutive scans for data reliability, and all spectra were obtained in positive ion mode. Moreover, the mass spectra were confirmed to be almost the same when comparing 30 or more ICR mouse models. A positive ion mass spectrum recorded from nonthermal AP plasma is shown in Supplementary Fig. 5. Despite the mass range being set to $m/z = 100$ to 1000 for measurements, strong ion signals were mostly observed under $m/z = 500$ from mouse hippocampal tissues. An examination of the MS spectra indicated the presence of more than 200 specimen-related ion species, and some of the detected ions were assigned to particular lipids and metabolites. A Q-Exactive hybrid quadrupole-Orbitrap (QE Orbitrap, Thermo Fisher Scientific, Germany) mass spectrometer was used to measure the exact mass of each peak in full scan mode, and each chemical formula was assigned using the XCalibur 3.0 software. With a strict comparison of the measured masses and the calculated chemical formulas, mass spectrum peaks observed in the mouse hippocampal tissue slices were identified as metabolites, lipids, and their derivatives, such as adenine, cholesterol, phosphocholine, and several fragments of glycerolipids and sphingolipids, as shown in Table 1. In addition, the assignments of cholesterol ($m/z = 385.346$) and adenine ($m/z = 136.061$), peaks were confirmed by a comparison of tandem MS data with the corresponding standard materials. The AP-nanoPALDI MS/MS spectra of the ion at $m/z = 385.346$ from the hippocampal tissue slices showed identical fragmentation patterns to a standard of cholesterol, as shown in Fig. 2b, c. In the same manner, the AP-nanoPALDI MS/MS spectra of the ion at $m/z = 136.061$ showed a fragmentation pattern identical to those of a standard of adenine (Fig. 3d, e). Unsaturated and saturated MAG ions ($m/z = 311.258$, $m/z = 313.273$, $m/z = 337.273$, $m/z = 339.289$, and $m/z = 341.305$) were reported to be well detectable in brain tissues in the positive ion mode[36–38] and were found to be precisely in agreement with chemical formulas, as shown in Table 1. In addition, by careful comparisons among the tandem MS data of the MAG ions, it was found that saturated MAG ions have distinctive fragmentation patterns while unsaturated MAG ions do not exhibit their distinctive fragmentation patterns (Supplementary Fig. 6). Several biomolecules including fragments

of sphingolipids were further assigned. Many of the ions assigned to $[M + H - H_2O]^+$ were in the form of $H_2O$ subtracted from the base ions $[M + H]^+$, which seems to be related to the experimental environment, such as humid specimen and open-air AP condition. It is important to note that signal intensities of the ions assigned to $[M + H - H_2O]^+$ proved sufficiently strong to construct MS imaging in general. In particular, only the cholesterol ion is seen as $[M−H]^+$ at $m/z = 385.346$ and $[M + H - H_2O]^+$ at $m/z = 369.352$. It is commonly reported that the spectra showed main ions at $m/z = 369.4$ and $m/z = 385.4$, respectively corresponding to $[M + H - H_2O]^+$ and $[M − H]^+$ for cholesterol in SIMS and laser ablation using MS experiments[37,39,40].

**MS imaging from live mouse hippocampal slice.** Using the AP-nanoPALDI MS system, including a scanning sample stage, MS images of a hippocampal tissue slice were obtained as shown in Fig. 3. The upper left and right images of Fig. 3 are optical microscopic images of a sample before and after MS analysis, respectively. When the analysis was completed, the area damaged by laser ablation was observed in all analysis regions. The lower images represent the spatial distribution of molecular ions in the hippocampal tissue slice. The ion images were generated with $433 \times 300$ pixels covering an area of $1800 \times 1500\,\mu m$. Consequently, mass spectra data were achieved from 129,900 different points and the corresponding x- and y-axial pixel sizes were 4.2 and 5 µm, respectively. Data acquisition took 70 s per scan line: 60 s to measure one x-axial scan line and a 10-s pause for preparing a measurement of the next line. Thus, the total data acquisition time for the MS imaging was 350 min (70 s per line scan × 300 lines).

Several identified ion images with sufficient signal intensity are shown in Fig. 3: ions at $m/z = 311.258$ of monoacylglycerol (MAG) (16:1), $m/z = 313.273$ of MAG (16:0), $m/z = 337.273$ of MAG (18:2), $m/z = 339.289$ of MAG (18:1), $m/z = 341.305$ of MAG (18:0), $m/z = 136.061$ of adenine, $m/z = 282.279$ of sphingosine, $m/z = 284.295$ of sphinganine (18:0), $m/z = 385.346$ of cholesterol, and $m/z = 548.539$ of ceramide (18:0). As shown in

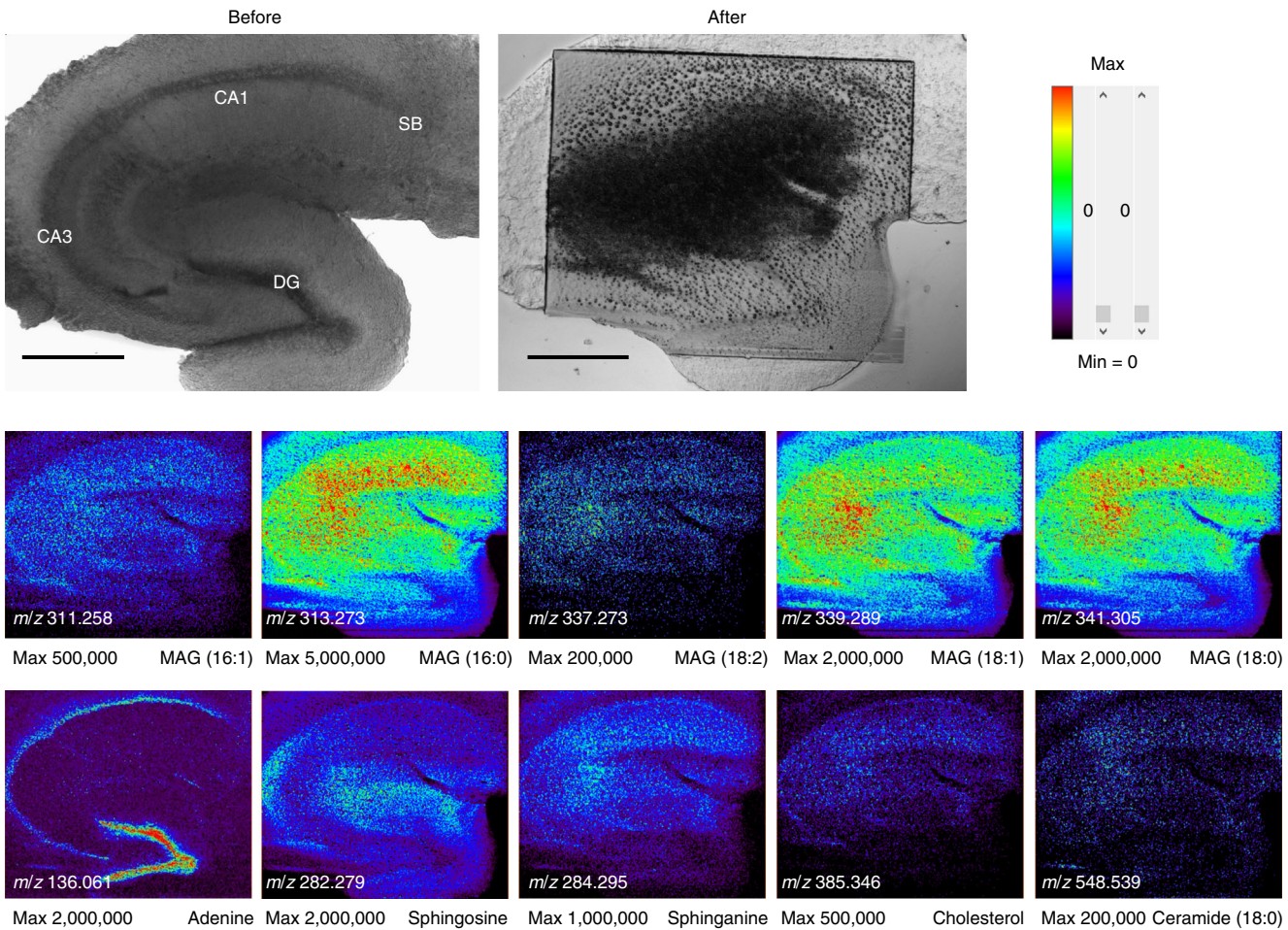

**Fig. 3** Optical images and mass spectrometric (MS) images of a mouse hippocampal tissue slice. The upper two optical images indicate differential interference contrast (DIC) images of a mouse hippocampal tissue slice. A clear distinction among sub-regions, such as cornu ammonis 1 (CA1), cornu ammonis 3 (CA3), dentate gyrus (DG), and subiculum (SB), could be observed. Before: before the MS analysis (the sample preparation was done). After: after MS analysis. Mass spectra data were achieved from 129,900 different points. Therefore, we could reconstruct MS data with spatial information and finally completed MS imaging. (MAG stands for monoacylglycerol in ion images.) Ion images were obtained from live hippocampal tissues of an adult mouse. Ion images were generated with 433 × 300 pixels covering an area of 1800 × 1500 μm, and the corresponding $x$-axial and $y$-axial pixel sizes were 4.2 and 5 μm, respectively. These ion images had $x$-axial and $y$-axial lateral resolutions of 5.1 and 5 μm, respectively (Fig. 5). Scale bars, 500 μm

Fig. 3, CA1, CA3, and DG regions were certainly distinguished in mouse hippocampal tissue. The identified ions exhibited mostly similar spatial distribution over the hippocampal tissue, except adenine ions. Interestingly, the adenine ion at $m/z = 136.061$ had a distinct spatial distribution. By comparison with the DIC image, it could be confirmed that the spatial distribution of adenine was highly correlated to soma (cell body) of neurons in the hippocampal tissue.

To monitor the sub-region of the hippocampal tissue in detail, we selected three sub-areas and analyzed them with a higher spatial resolution. The AP-nanoPALDI MS system has the advantage of precisely selecting a small region due to the microscope-based MS analysis system. Since the sampling procedure, including desorption and ionization, occurred on the stage of the microscope, the analysis region could be directly defined by $x$- and $y$-positioning of the stage while looking through the microscope. As shown in Fig. 4, we can compare MS images of specific molecular ions for the selected region. The ion images were generated with 433 × 100 pixels covering an area of 600 × 500 μm, and total data acquisition time was 117 min (70 s per line scan × 100 lines). Therefore, the corresponding $x$-axial pixel size of 1.4 μm was achieved at a sample moving velocity of 10 μm s$^{-1}$, and the $y$-axial pixel size was 5 μm (see Supplementary

Note 2 for a detailed description). Since the analysis region was decreased, the $x$-axial pixel size of the ion image could be increased with a limit, as discussed in the Discussion section, and higher-resolution MS imaging could be obtained. The $y$-axis pixel size could be improved by reducing the interval between adjacent $x$-lines; however, the $y$-axial pixel size was kept at 5 μm to avoid the increase of the analysis time. Higher-spatial-resolution MS images of CA1, CA3, and DG areas of the adult mouse hippocampus were obtained, as shown in Fig. 4, and details of molecular differences between the apical and the basal dendrite regions monitored by MS imaging are discussed in the Discussion section below.

**Subcellular spatial resolution of the AP-nanoPALDI method.** To investigate the spatial resolution of the AP-nanoPALDI method, the resolution test specimen was designed using a gold transmission electron microscopy (TEM) grid with 400 mesh (see Methods section) so that the resultant MS imaging would have several straight edges as shown in Fig. 5. Figure 5a, b shows the ion images of the MAG (16:0) at $m/z = 313.273$ from the hippocampal tissue slice when the sample moving velocities for raster scanning were set differently to 10 and 30 μm s$^{-1}$,

respectively. The $x$-axial lateral resolution of the MAG (16:0) ion image has been derived from the average of the three neighboring scan lines, as shown in Fig. 5c, d and Supplementary Fig. 7. The distance across which the signal change from 16 to 84% (or 84 to 16%) of the maximum indicated an $x$-axial lateral resolution[41,42]. As a result, $x$-axial lateral resolutions of these ion images of $2.9 \pm 0.6$ and $5.1 \pm 1.1 \, \mu m$ were achieved (means $\pm$ SDs from 10

distinct edges are shown in both cases of Fig. 5c, d) when the sample moving velocities were 10 and 30 $\mu m \, s^{-1}$, respectively. We also confirmed similar micrometer lateral resolution from MS imaging of the caudal fin of zebrafish (Supplementary Fig. 8).

To take MS images with high spatial resolution, the uniform and strong desorption behavior used by a small-sized focused laser beam and uniformly distributed AuNRs inside the whole

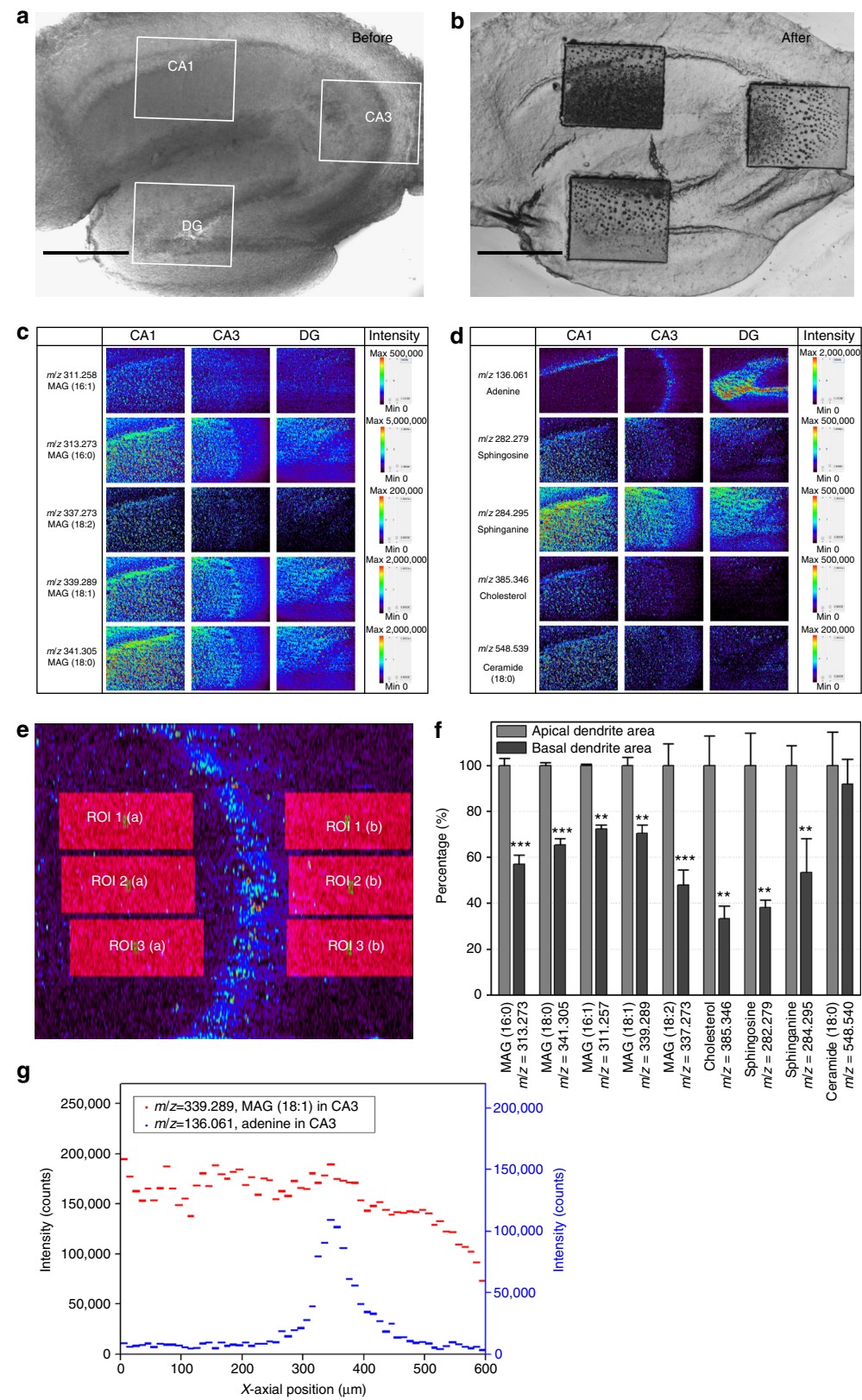

tissue specimen were necessary. Thus, we needed to check the morphology of the hippocampal tissue after MS analysis. In order to unambiguously measure the morphological changes of the focused laser-treated biological specimen in a label-free fashion, we employed coherent anti-stokes Raman scattering (CARS) microscopy[43]. Figure 6a–e showed the 3D reconstructed z-stack CARS images of the sample collected after the analysis process based on this scheme. We obtained 95 CARS images of the CH stretching region (2750–2960 cm$^{-1}$) over the arbitrarily chosen mass-spectroscopically scanned position with a z-axial stepping interval of 0.5 μm and merged them as one feature using 3D visualization software (Amira 5.3.3, Visualization Sciences Group, France). More specifically, the 2D CARS images composed of 512 × 512 pixels were z-directionally stacked in 95 layers, covering a total volume of 635 × 635 × 47 μm$^3$. The images of Fig. 6a–c, e achieved from the same target were reformatted with regards to the different view angles and magnifications for better understanding. Since the spot size (about 1.2 μm) of the focused lasers was smaller than 5 μm of the spacing between adjacent scanning lines, the tracks of laser scanning could be clearly visualized, as shown in Fig. 6b, c. In addition, the CARS intensity profile represented in Fig. 6d showed the length of one trench by the laser scanning was 5 μm; the desorption behavior occurred in a quite uniform and stable manner over the whole analysis area. As depicted at the edge of the analysis region (Fig. 6e), the desorbed area was shown to have a very steep morphology, and a mass volume with 35-μm thickness from the surface of the mouse hippocampal tissue (~74% of the total tissue layer) was actually desorbed and used for MS analysis. This clearly shows that this AP-nanoPALDI MS process is not a surface or near-surface analysis but a bulk analysis of materials, in contrast to DESI.

To measure the dimension of the linear crater generated by the focused laser beam on the hippocampal tissue slice during raster scanning, 3D confocal imaging of the individual scan lines ablated by the laser beam were obtained from the confocal laser scanning microscope (LSM-700, Carl Zeiss, Germany), as shown in Fig. 6f. At the bottom of the crater, a plateau was clearly observed. The bottom and top widths of the ablated linear crater were estimated to be about 3 and 5.5 μm from the average of thirty points, respectively. When the positioning stage moved at a constant velocity of 10 μm s$^{-1}$, the hippocampal tissue was exposed to fs laser pulses with pulse energy of 2 nJ and a pulse repetition rate of 75 MHz. The linear crater observed by helium ion microscopy (HIM, Orion NanoFab, Carl Zeiss, USA) was consistently quite steep and clear, as shown in Fig. 6g.

## Discussion

In many cases of AP ionization methods for mass spectrometry, a sampling region was heated using the heated gas stream or the heated sample substrate during the sampling procedure for an improvement of desorption and ionization of volatile organic compounds[26,44–46]. However, since we could not additionally heat the living tissue specimen to avoid heat damage, we employed fs laser oscillators and nonthermal AP plasmas simultaneously as AP desorption/ionization sources.

The application of AuNRs to the live tissue slice was the key to successful desorption by the fs NIR laser oscillator. The AuNRs with aspect ratio 4.0 ($\lambda_{max} = 800$ nm) could absorb the NIR light and rapidly convert the absorbed energy into thermal energy[28,47,48]. Through the uptake of AuNRs by tissue membranes, AuNRs inside live tissue played a critical role as hot spots interacting with NIR light, which resulted in a significantly enhanced desorption procedure and higher mass spectra intensity. In addition, since the repetition rate of the fs laser oscillator was 75 MHz, which corresponded to a train of pulses 13 ns apart, the tissue specimen was continuously irradiated and desorbed by an enormous number of laser shots during uniform motion. Therefore, the desorption induced by the fs NIR laser oscillator was observed to be uniform and stable, and its profile was sharp and steep, as shown in Fig. 6, resulting in MS imaging with subcellular spatial resolution.

In order to achieve high-quality MS imaging, the uniform distribution of AuNRs in the live tissue specimen was the most important consideration. Thus, we used mPEG-modified gold nanorods (mPEG-AuNRs) for uniform distribution of AuNRs within the tissues. The live tissue slices were submerged and incubated with mPEG-AuNRs containing artificial cerebral spinal fluid (ACSF) for 1 h. After incubation, the tissues were washed with ACSF 10 times only, to keep AuNRs embedded inside the tissue. TEM images were obtained to investigate the actual distribution of AuNRs inside the tissue, as shown in Supplementary Fig. 9. TEM images confirmed that mPEG-AuNRs were fairly evenly distributed throughout the tissue. Because the particle size of mPEG-AuNRs was very small compared to the size of the focused laser beam, it was not necessary that the nanoparticles were ideally evenly distributed in the tissue. As a result, the desorption induced by the fs NIR laser oscillator was shown to be uniform and stable owing to the efficient assistance of AuNRs over the whole analysis region, as shown in Fig. 6a–e. In spite of the small beam size of the laser (1.2-μm diameter), the desorbed tissue thickness was 35 μm, as shown in Fig. 6e. It was noticed that this high aspect ratio of the laser-ablated crater was detrimental to the spatial resolution in the depth direction. The tissue slices, pseudo-two-dimensional organotypic models, could be normally analyzed with the AP-nanoPALDI method without difficulty. However, because of a limitation of the depth resolution in the AP-nanoPALDI method, careful analysis and interpretation were required when analyzing inhomogeneous tissue specimens in the depth direction, such as cancer tissues. Without AuNRs treatment for MS imaging, no noticeable mass ion peaks of biomolecules were observed.

In contrast to the MALDI and surface-assisted laser desorption ionization (SALDI)[49,50] methods, a separate ionization source, a nonthermal AP plasma jet, served to perform a subsequent post-ionization process in the AP-nanoPALDI method. Since the AP plasma jet employed in our system had non-thermal plasmas behaviors, even if they contained lots of charged species including

**Fig. 4** Spatial distribution of detected ions with sub-regions of a mouse hippocampal tissue by MS imaging. Two optical images indicated three analysis regions of CA1, CA3, and DG; **a** before the MS analysis and **b** after MS analysis. Scale bars, 500 μm. **c, d** identified ion images were compared with CA1, CA3, and DG. Ion images were generated with 433 × 100 pixels covering an area of 600 × 500 μm, and the corresponding x-axial and y-axial pixel sizes were 1.4 and 5 μm, respectively. According to experiment, the ion images had x-axial and y-axial lateral resolutions of 2.9 and 5 μm, respectively (Fig. 5). **e** Six identical ROIs were assigned to each side based on adenine boundary of CA3 using the function of ROI analysis in BioMap software. The three left areas indicate the apical dendrite areas and the three right areas indicate the basal dendrite areas. **f** Relative proportions of detected ions, where the intensities of apical dendritic populated areas were normalized to 100% (raw data is shown in Supplementary Table 1). Shown are means ± SDs from nine independent ions in each of the three regions. Asterisks indicated P values. \*\*P < 0.01; \*\*\*P < 0.001. **g** Distribution of ion intensities of MAG (18:1) in **c**, and adenine in **d**, along x-axial position. The values of graph represented averages of forty y-axial values per each x-position (middle 40 lines of 100 lines in y-direction) and also averages per 10 μm along the x-direction, from 0 to 600 μm

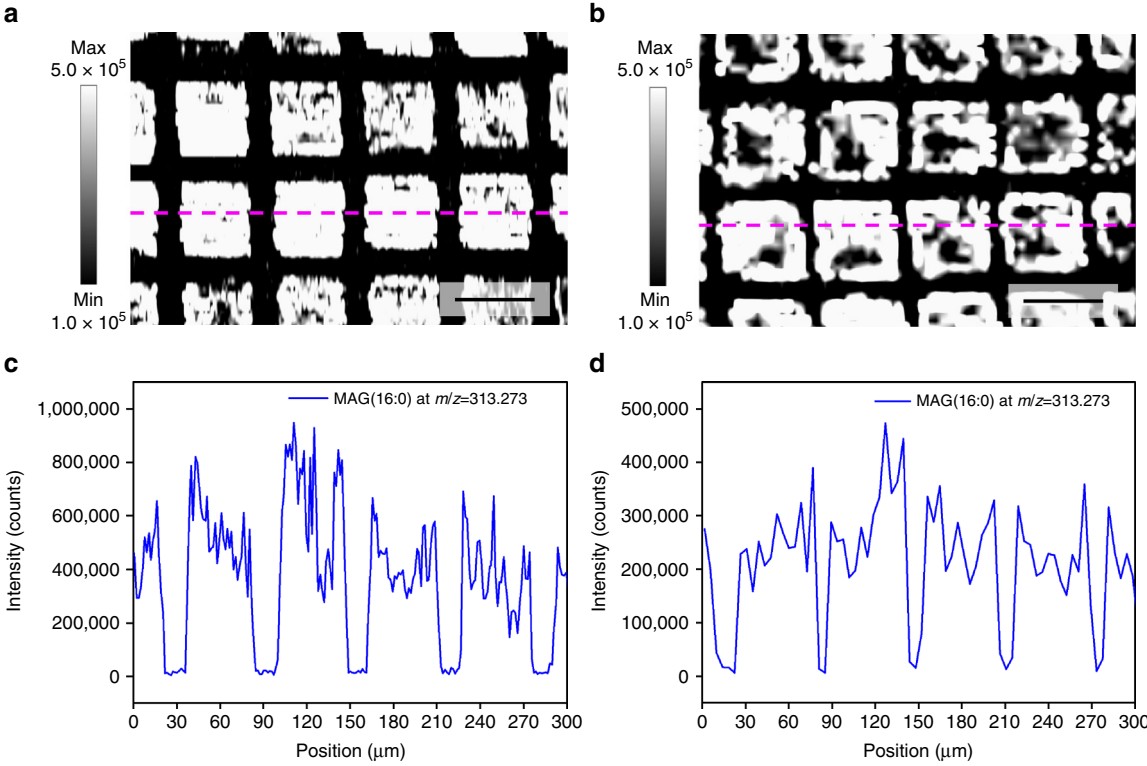

**Fig. 5** The *x*-axial lateral resolution test of AP-nanoPALDI imaging in a hippocampal specimen on 400 mesh grid. Ion images for MAG (16:0) ion at *m/z* = 313.273 were taken during the analysis of the hippocampal tissue slice (the area of 300 × 200 μm) on a TEM grid with 400 mesh when the sample moving velocities were set to **a** 10 μm s$^{-1}$ and **b** 30 μm s$^{-1}$, respectively. Scale bars, 50 μm. The line scans were obtained from the MAG (16:0) ion image at the average of the three neighboring scan lines (magenta dashed line). The resulting line scan (**c**) corresponds with ion image (**a**), and line scan (**d**), corresponds with ion image (**b**). As a result, the *x*-axial lateral resolutions achieved were 2.9 ± 0.6 and 5.1 ± 1.1 μm when the sample moving velocities were 10 and 30 μm s$^{-1}$, respectively. They represented means ± SDs from 10 distinct edges in each case as shown in **c** and **d**

electrons, ions, and excited species, the temperature of these plasmas was low enough to avoid harming the live biological sample[51,52]. Moreover, the generated plasma jet could not directly contact the sample due to the suctioned air flowing in the configuration of the sampling equipment on the sample stage, as shown in Supplementary Fig. 2c,d. This series of desorption and post-ionization was very effective for analysis of live biological samples under open-air AP condition.

Owing to the application of both the nanoparticles and the separate post-ionization source, pulse energy of the laser could be remarkably reduced by comparison with a use of the laser source alone. The lowering of the pulse energy not only reduced thermal damage to the specimen but also constricted scattering behaviors of desorbed materials caused by laser bombardment, resulting in a fairly sharp and clear desorption pattern. This is the reason why we use a cost effective and simple structured fs laser oscillator with pulse energy far lower than that using an fs laser amplifier (for instance, 2–4 nJ vs. 150–800 μJ) as a desorption source. Therefore, despite continuous irradiation with a large number of laser shots, no damage caused by the burn around the linear craters was found, as shown in Fig. 6g.

In addition, the proposed AP-nanoPALDI method can analyze fresh tissue slices with humid in open-air AP and ambient temperature conditions. SIMS specimens were commonly prepared by cryosection and drying so that distortions of lipid imaging were quite often noticed[53–55]. The distortion of the original cholesterol distribution could be clearly observed by an HIM image and a cholesterol SIMS image (TOF.SIMS V, IONTOF, Germany), shown in Supplementary Fig. 10. In the HIM image in Supplementary Fig. 10a, segregated cholesterol crystals formed

during the SIMS sample preparation procedure could be clearly observed. Therefore, the SIMS image in Supplementary Fig. 10b might be different from the original cholesterol distribution. Cholesterol images in Figs. 3 and 4 taken from fresh tissue slices were free from any distortions due to sample cryosection and drying while keeping the subcellular spatial resolution.

Neurons of the central nervous system exhibit immensely diverse dendritic arbor structure, which is closely related to neuronal connectivity[56]. In the hippocampal formation, pyramidal cells in the cornu ammonis (CA) and granule cells in the dentate gyrus (DG) are known as two principal neuronal types, exhibiting distinct dendritic arbor structures[57]. As shown in Figs. 3 and 4, fragments of glycerolipid and sphingolipid, and cholesterol ions were observed with higher signal intensities in the CA1 and CA3 region than those in the DG area. In contrast, the adenine ions at *m/z* = 136.061 were observed to be concentrated in the DG area. This difference in ion image of adenines between CA and DG regions showed that two neuronal types, pyramidal cells and dentate granule cells, have different distributions and cell phenotypes in mouse hippocampal tissues. Actually, the granule cell bodies were known to be tightly packed and to have less glial sheath interposed between cells in the dentate gyrus[58]. Thus, adenine ions, which were commonly observed in nucleus, were concentrated in the DG.

In a similar manner, pyramidal cells in the CA1 and CA3 region structurally have two different dendrites: a long and thick apical dendrite and several multiple basal dendrites that emerge from the apex and base of the soma, respectively[56]. The apical and basal dendrites were known to be different in many properties, such as size, geometry, electrical conduction, and

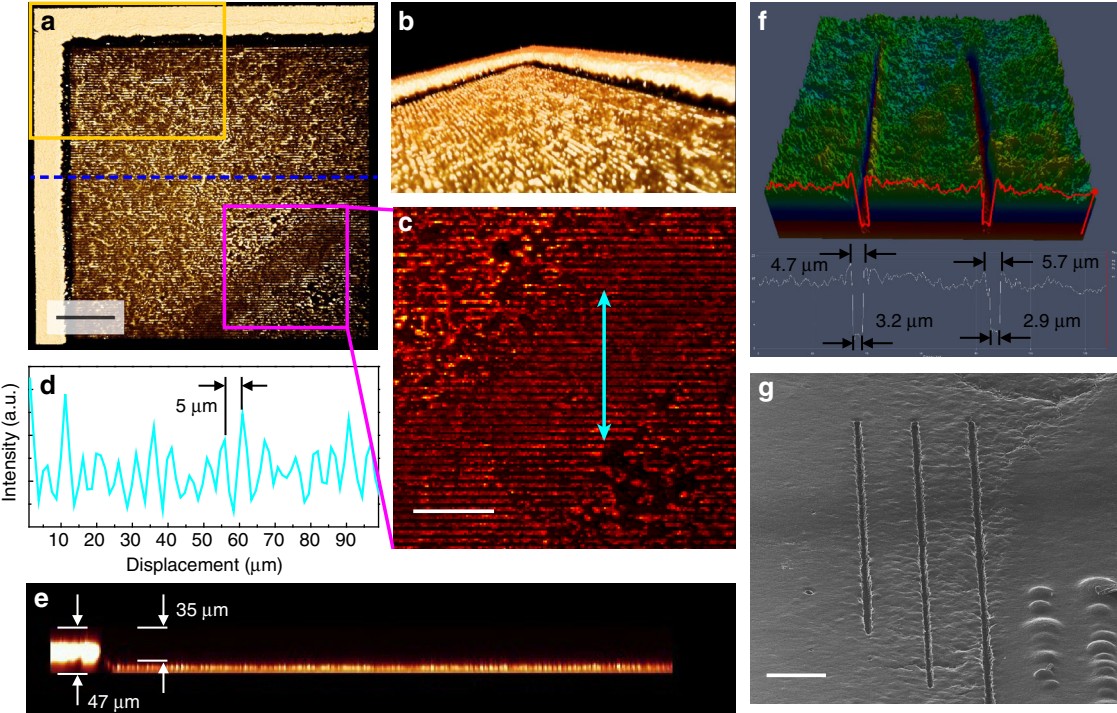

**Fig. 6** Morphological changes of hippocampal tissue specimen after AP-nanoPALDI process. **a–e** Coherent anti-stokes Raman scattering (CARS) images taken with regards to the vibrational modes from 2750 to 2950 cm$^{-1}$ after AP-nanoPALDI MS analysis; **a** top view, **b** side view of the orange rectangle in **a**, and **c** zoomed-in view of the magenta square in **a**. **d** The CARS intensity profile along the double sided arrows in **c**, shows the length of one trench was very close to 5 μm. **e** Cross-sectional view across the horizontally drawn blue dotted line in **a**. **f** Line patterns profile by 3D confocal imaging and, **g** helium ion microscopy (HIM) imaging of individual line patterns ablated by AP-nanoPALDI. All the images were taken after the hippocampal tissue was dried. Scale bars, **a** 100 μm, **c** 50 μm, and **g** 50 μm

responsiveness to neurotrophic factors or guidance molecules[56,59,60]. Moreover, while morphological difference between apical and basal dendrites has been studied extensively[56–60], the molecular composition difference between them has not been, due to the lack of reliable molecular markers or adequate molecular composition analysis methods. To investigate the molecular difference between the apical and basal dendrites of pyramidal neurons at the tissue level, the sub-regions of the hippocampal tissue were analyzed as shown in Fig. 4c, d. In particular, ion images in the CA3 can be clearly divided on both sides of the adenine location, which represented the sites of cell bodies (ion image at $m/z = 136.061$), as can be seen in Fig. 4c–e. The left side of the ion images in the CA3 represented a densely populated area of the apical dendrites, and the right side represented a densely populated area of the basal dendrites in the hippocampal tissue.

Using the ROI analysis function of BioMap software, the identical areas of ROI were assigned to both left and right sides and their intensities of mass spectra of monoacylglycerols, sphingosine, sphinganine, cholesterol, and ceramide ions compared in detail (Fig. 4e, f and Supplementary Table 1). The signal intensities of MAG ions (at $m/z = 311.258$, 313.273, 337.273, 339.289, and 341.305) in the basal dendrite side were 48–72% of those in the apical dendrite side. In contrast, the ceramide ion at $m/z = 548.539$ was observed to be similar on both sides. The high-resolution MS imaging from the AP-nanoPALDI method clearly revealed that MAGs are distributed 1.4–2.1 times more in the apical dendrite populated area than the basal dendrite populated area, while the ceramide at $m/z = 548.539$ is relatively evenly distributed between both dendritic areas. Noticeably, the cholesterol at $m/z = 385.346$ in the apical dendrite populated area had almost three-times-higher signal intensities than in the basal

dendrite populated area. With these high-resolution MS images, we found that the apical dendrites contained more monoacylglycerols, sphingosine, sphinganine, and cholesterol than the basal dendrites, while the ceramide seems to have even distributions in the two dendritic structures in the CA3 of the live hippocampus tissue. This similar difference between the apical dendrite region and the basal dendrite region can be also observed in CA1 in Fig. 4c, d. Details of the AP-nanoPALDI MS imaging in Fig. 4c, d of the apical dendrite region of CA3 may indicate the mossy fiber bundles from the DG granule cells extending parallel to the pyramidal cell layer in the stratum lucidum[61,62].

In the proximal apical dendrite region of the CA3, an area of 100-μm width with lower intensities of MAG can be noticed along the pyramidal soma line in Fig. 4c. In the CA3 area, mossy fibers run in two main bundles, the main suprapyramidal projection in stratum lucidum and the intra- and infra-pyramidal projection that runs first within the proximal extent of the stratum pyramidale and stratum oriens, to cross over to the stratum lucidum in the CA3[61,62]. CA3 pyramidal neurons make synaptic contacts with mossy fiber butones via a specialized complex of clustered spines called thorny excrescences. In the distal CA3 neurons close to the CA1, the thorny excrescences are found primarily on the apical dendrite at distances of 10–120 μm from the soma[63]. In Fig. 4g, the line profiles of adenine and MAG (18:1) from Fig. 4c, d are given, where the adenine profile clearly shows the position of the pyramidal soma line. The MAG (18:1) profile shows a low intensity area of 100-μm width in the apical dendrite region next to the soma line, which is consistent with the lamellar structure due to the mossy fiber projection from granule cells and the thorny excrescence from CA3 neurons. For cholesterol and ceramide (18:0) images, the similar distributions are

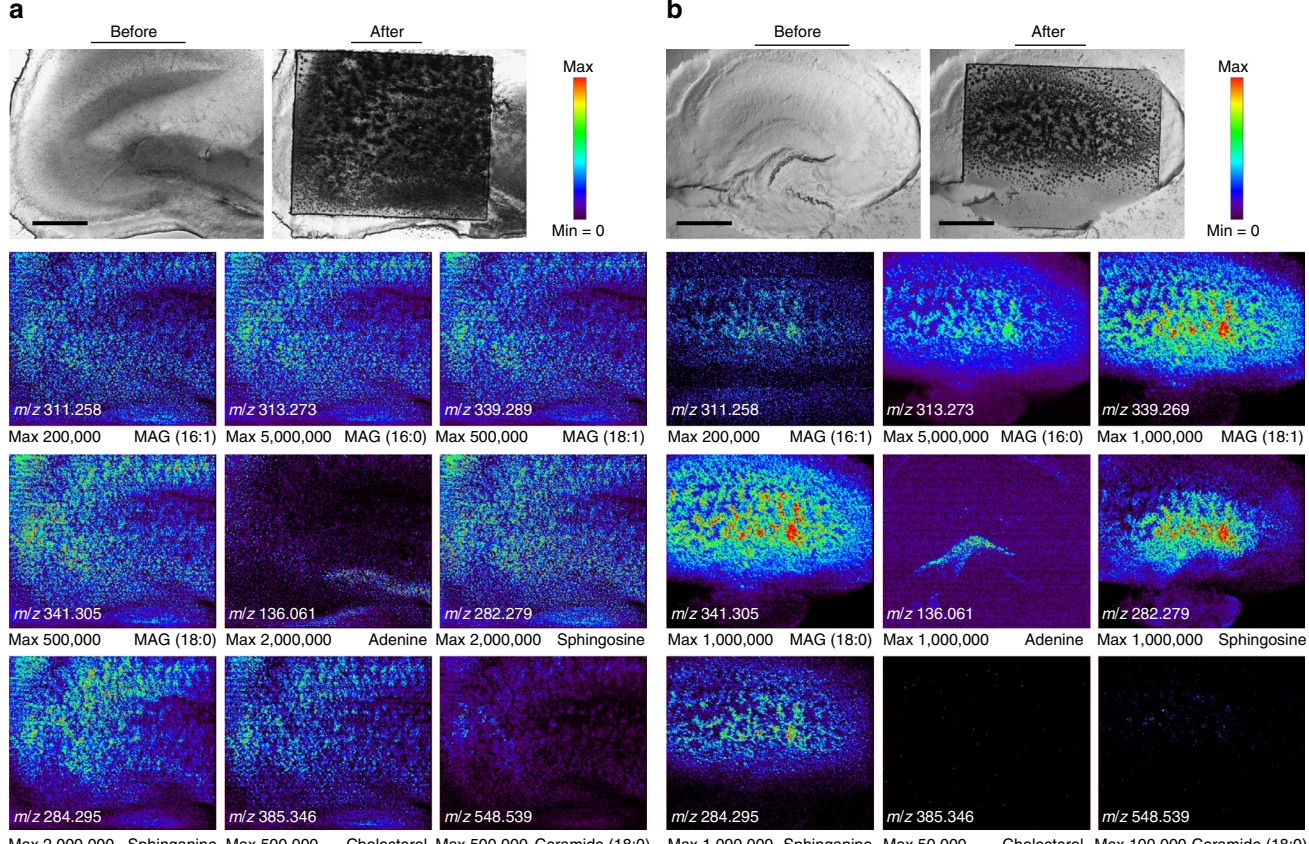

**Fig. 7** Ion images obtained for investigation of drug influence. **a** Normal hippocampal tissue and **b** a methyl β-cyclodextrin (mβCD)-treated hippocampal tissue of an adult mouse. All experimental and MS analytic conditions were the same as those in Fig. 3. Because the ions of MAG (18:2) shown in Figs. 3 and 4 have relatively low signal intensity as well as similar spatial distribution of other MAG ions, the ion image of MAG (18:2) at $m/z = 337.273$ was not shown here for nine image layouts in a square shape. Scale bars, 500 μm

not observed. This observation indicates that AP-nanoPALDI can reveal the molecular composition difference between the mossy fiber projection and other apical dendrite regions. In the CA1 region, no similar features are observed in the apical dendrite region, as in Fig. 4c, d, since the mossy fiber projection does not reach CA1.

AP-nanoPALDI MS images from live tissue slices can provide plenty of spatial information for metabolites, which could be useful for tissue-based drug screening. In order to observe tangible drug influences of hippocampal tissues, cholesterol depletion treated by methyl β-cyclodextrin (mβCD) was investigated by AP-nanoPALDI MS imaging. Aerated mβCD-treated mouse hippocampal tissues were dipped in ACSF containing mβCD (25 mg ml$^{-1}$) for 6 h. More detailed descriptions of mβCD-treated hippocampal tissue slices are given in the Methods section below. The ion images were achieved from two different hippocampal tissues of the same animal: normal hippocampal tissue (Fig. 7a) and mβCD treated hippocampal tissue (Fig. 7b). It is noteworthy that these two hippocampal tissue slices were serial sections from the same ICR mouse in spite of difference in shape, because the specimens were prepared without being frozen. For the mβCD-treated hippocampal tissue, the ion at $m/z = 385.346$ of a cholesterol peak was suppressed and disappeared in the entire analysis region as shown in Fig. 7b. Interestingly, the ceramide ion at $m/z = 548.539$ was observed to be also quite suppressed. The other identified ions were observed to be slightly lower than the control sample. Consistent results to former reports for mβCD-induced cholesterol depletion behavior was found in these tissue slices[64,65]. We believe that AP-nanoPALDI MS imaging can be

applied for tissue-based drug screening[66,67], which will provide higher reliability than the cell-based drug screening with sacrifice of fewer animals.

## Methods

**Mass analyzer.** The mass analyzer employed was a Q-Exactive hybrid quadrupole-Orbitrap mass spectrometer (Thermo Fisher Scientific)[68]. MS parameters for the full MS scan were as follows: positive ion mode, mass resolution of 35,000 FWHM, mass range of 100–1000 $m/z$, 1 microscan, maximum injection time of 100 ms, automatic gain control (AGC) target of $10^6$, capillary temperature of 350 °C. Parameters for the tandem MS scan were as follows: mass resolution of 17,500 FWHM, isolation window of 0.4 $m/z$, high-energy collisional dissociation (HCD) with normalized collision energy (NCE) of 20–80, AGC target of $2.0 \times 10^5$, maximum injection time of 100 ms, underfill ratio of 1.0%, exclude isotopes "on", and dynamic exclusion 5.0 s. Using Inclusion and Exclusion Indexes, the particular ion peaks were analyzed with MS/MS.

**Near infrared fs lasers.** An fs laser oscillator was used as an AP desorption source of the AP-nanoPALDI MS system. The fs laser consisted of a mode-locked Ti-Sapphire oscillator (Synergy 20, Femtolasers, Austria), which gives 75 MHz, <20 fs pulses at 802 nm with a maximum single pulse energy of 5 nJ. The energy of the fs laser system was controlled using a graduated neutral density (ND) filter prior to the rear port of the inverted optical microscope, which served as an analysis stage for samples. An fs laser beam was integrated with the inverted microscope by placing a dichroic beam splitter (FF720-SDi01, Semrock, USA) prior to the input aperture of the objective lens of the microscope. The final pulse energy applied to the samples was about 2 nJ at 802 nm.

**Atmospheric pressure plasma jet.** The AP plasma jet used the conventional double electrode configuration for a dielectric barrier discharge (DBD) jet[69]. Nonthermal plasma jets were generated in a quartz tube with an inner diameter of 2 mm and an outer diameter of 3 mm. The 6-mm-wide electrodes were made of copper tape wrapping the quartz tube, and the gap between the inner edges was

7 mm. The ground electrode was on the upstream side; the powered electrode was on the downstream side and 5 mm apart from the tube orifice. It is a typical dielectric barrier discharge device with a small discharge current. The sinusoidal voltage, with a peak value of 5 kV and a frequency of 27 kHz, was applied to the plasma jet device. A high-purity helium gas (HP grade; 99.999%) with a gas flow rate of 0.5 slm was used as a discharge gas. The plasma device was positioned toward the sample with 30° capillary angle of incidence.

**Airflow-assisted ion transport system**. The customized chamber was installed to the MS inlet to allow installation of the transfer tubing and connection to a pump. Stainless steel tubing (i.d. 4.57 mm, o.d. 6.35 mm, length 120 mm) was used as an ion transfer tube with a 30° downward bend to be close to the stage. The 60° curved glass capillary (i.d. 2 mm, o.d. 2.5 mm, length 20 mm) was connected to the transfer tube for effective collection of molecules and ions from the sample. The dry pump (WOB-L pump 2546, Welch Vacuum) was used for providing the auxiliary airflow to improve ion transport inside the transfer tubing. The pump was connected to the side hole of the chamber via silicone tubing with o.d. 6.35 mm (1/4 inch). A 30 l min$^{-1}$ flow rate was provided in the transfer tube.

**Programmable auto-scanning stage**. Mass spectrometric data were acquired as a bundle of line scans. In the programmable motorized X–Y scanning stage (AS-MIX73-C, iNexus, South Korea), the stepper motors were externally controlled via both a joystick and a motion control program. All scanning parameters, such as X-Y coordinates, scan speed, scan direction, interruption time, and number of scan, were programmable using the customized stage control software. In addition, its motion control program relayed logic signals to both the scanning stage and the mass analyzer simultaneously. This allowed the scanning process to be synchronized with the MS data acquisition with good accuracy. The sample stage was line scanned in the x direction, and fed in the y direction for MS imaging. Each single line scan along the x direction was saved as one data file. Using the sequence mode of the data acquisition program in this mass analyzer, several hundred data files could be achieved in one experiment. The data files of multiple scan lines from the analytic area must be assembled in one data file for the MS image plotted against the x and y coordinates for a set of line scans in the x direction and fed in the y direction.

**Software for MSI**. FireFly 2.2.00 for Thermo (Prosolia, USA) was used for extracting and combining data sets into MS imaging file formats compatible with BioMAP, a third-party image visualization and processing software program (Novartis Institutes for BioMedical Research, http://www.maldi-msi.org). BioMAP was used for all image reconstruction and display of ion distribution images in this work. Intensity maps were displayed by selecting intensity range corresponding to the m/z value of the analyte.

**mPEG-AuNRs preparation**. First, CTAB-AuNRs ($\lambda_{max} = 800$ nm) were synthesized by following the established method[70]. In the first step, seed solution was prepared by mixing 9.75 ml of CTAB (0.1 M) with 0.25 ml of HAuCl$_4$ (0.01 M) and 0.6 ml of a freshly prepared ice-cold solution of NaBH$_4$ (0.01 M). The solution was stirred vigorously for 1–2 min and then kept at 28 °C for 3 h. In the second step, growth solution was prepared by adding 3 ml of AgNO$_3$ (0.01 M), 20 ml of HAuCl$_4$ (0.01 M) and 3.2 ml of freshly prepared ascorbic acid (0.01 M) to 475 ml of CTAB (0.1 M). To this solution, 3.2 ml of seed solution was added and the reaction mixture was subjected to gentle shaking for a few seconds and was then kept undisturbed for at least 6 h at 28 °C. The color of solution changed to purple with the formation of AuNRs. The aspect ratio of AuNRs was estimated be ~3.4, with an average length of 41.9 (±4.4) nm and width of 12.2 (±1.3) nm with a localized surface plasmon resonance peak ($\lambda_{max}$) at around 765 nm. To prepare mPEG-modified AuNRs, 20 ml of CTAB-AuNRs solution was centrifuged at 15,000 rpm for 15 min. After discarding the supernatant, AuNRs were re-dispersed in 10 ml of distilled water (DW). Into this solution, mPEG-SH (MW 6000, 2.0 mg) was added, followed by the addition of 10 ml ethanol. The solution was stirred vigorously for a few seconds and maintained with gentle shaking for 12 h. The unreacted mPEG was removed by centrifugation at 12,000 rpm for 15 min, and the mPEG-AuNRs were re-suspended in DW (10 ml). The optical characterizations of fabricated mPEG-AuNRs are shown in Supplementary Fig. 4.

**Preparation of hippocampal brain slices**. Since the obvious aim in this study was to analyze live biological tissues, the sample preparation method was similar to that of multielectrode arrays (MEAs) studies[34]. Male 6-week-old ICR mice, were purchased from Koatech (Pyeongtaek, South Korea) and housed in pathogen-free animal facilities. All experiments were approved and performed in accordance with the Daegu Gyeongbuk Institute of Science & Technology (DGIST, South Korea) for animal use and care guidelines. After sacrificing and brain extraction, the hippocampus was isolated and transversely sliced at ~200 μm thickness with a tissue chopper (McLlwain tissue chopper, Cavey Laboratory Engineering, UK). Slices were aerated with oxygenated sucrose artificial cerebral spinal fluid (sACSF)

containing 124 mM NaCl, 2.5 mM KCl, 1.25 mM KH$_2$PO$_4$, 26 mM NaHCO$_3$, 2 mM MgSO$_4$, 2.5 mM CaCl$_2$, 10 mM D-glucose, and 4 mM D-sucrose bubbled with 95% O$_2$/5% CO$_2$ using an aquarium bubbler at 32 °C for 2 h.

For cholesterol removal, slices were washed with sucrose-free ACSF and incubated for 6 h at 32 °C with sucrose-free ACSF in the absence (control slices) or presence (treated slices) of 25 mg ml$^{-1}$ methyl β-cyclodextrin (mβCD, Sigma-Aldrich, USA). After 6 h aeration, the slices were submerged with 5 ml of ACSF solution with AuNRs (Optical density 10.0 at 800 nm, 200 μl). After 1 h incubation with the AuNRs, the hippocampal slices were washed 10 times and placed on 0.1% polyethylenimine (PEI, in 25 mM borate buffer)-coated slides to facilitate tissue adhesion. Since these tissue samples do not need to be completely dried, the analysis can be started within 20 min.

**Preparation of spatial resolution test specimen**. A hippocampal tissue slice treated with mPEG-AuNRs was placed onto a 400 rectangular mesh grid (G400, GilderGrids, UK) on the glass slide. The 400 rectangular mesh grid employed in this test had a pitch of 62 μm, mesh openings of 37 × 37 μm$^2$ separated by 25-μm bars, and a 20-μm thickness. As the gold mesh grid played the role of a mask for preventing laser ablation, the resultant MS imaging presented distinct border patterns.

**3D CARS imaging**. After the AP-MS imaging, the sample was fixed with 4% paraformaldehyde solution overnight at 4 °C, and then subjected to a rigorous cleansing process using the water and ethyl alcohol solutions before being mounted to the imaging chamber. Optical images were captured by a home-built CARS microscope constructed with the combinatorial usages of the ultrafast pulse (~120 fs) laser sources (Insight deepsee dual, Spectra Physics, USA) and the x–y directional galvanometric scanner (FV-1000, Olympus, Japan) equipped to the inverted microscopic system (IX-83, Olympus, Japan) with a ×20 objective lens (UPLSAPO20X, NA: 0.75)[43]. The lipid-sensitive chemical contrast was created by mixing 802 nm pump and 1041 nm of Stokes pulse trains (80 MHz) tuned to the central vibrational mode of 2860 cm$^{-1}$. The acquired CARS images were critically analyzed and processed using Olympus Fluoview 1.7a software. The z-directionally scanned images were stacked with an axial step size of 0.5 μm to produce 3D images as well as cross-sectional views. The stacked images were then rendered as a 3D structure using Amira® 5.3.3 image-analysis software (Visualization Sciences Group, France) for more graphical effect.

**3D confocal laser scanning imaging**. After the laser desorption procedure, the tissue sample was fixed with 4% paraformaldehyde solution using the same method used for CARS. Optical images were captured by a confocal laser scanning microscope, a Zeiss AxioImager Z2m (LSM-700, Carl Zeiss, Germany) equipped with a laser scanning unit. Optical train was equipped with ×100/0.8 HD Zeiss EC Epiplan objective. Illumination was provided by a 5-mW solid-state laser generating a monochromatic light with 405-nm wavelength. The z-directionally scanned images were stacked with an axial step size of 0.7 μm to produce 3D images as well as cross-sectional views. Image documentation and analysis were done using LSM Software (ZEN 2010M).

**Helium ion microscopy imaging**. Helium ion microscopy (HIM) measurements were performed with an Orion NanoFab (Carl Zeiss, USA) with beam energy of 35 keV and a 0.05–0.5-pA probe current. Notably, no conductive coating was applied to the samples prior to imaging. Setting information for HIM imaging of Fig. 6g was as follows: tilt of 55°, beam current of 0.244 pA, scan number averages of 128, working distance of 10.92 mm, gas field ion sources (GFIS) aperture size of 20 μm, scan dwell time of 0.5 μs, and GFIS field of view 350 μm.

**TEM imaging**. Primary fixation was accomplished at 4 °C for 4 h using 2% paraformaldehyde and 2% glutaraldehyde in 0.05 M sodium cacodylate buffer at pH 7.2. For post fixation, 1% osmium tetroxide in 0.05 M sodium cacodylate buffer was used at 4 °C for 2 h. En bloc staining was performed at 4 °C for 30 min with 0.5% uranyl acetate. The samples were dehydrated at room temperature for 10 min at each step with 30%, 50%, 70%, 80%, 90%, 100%, 100%, and 100% ethanol. Transition was performed twice at room temperature for 15 min with 100% propylene oxide. Infiltration was performed using Spurr's resin. Polymerization was performed at 70 °C for 24 h. The samples were sectioned to a thickness of 80–120 nm by Ultramicrotome (MT-X, RMC, USA), and then stained with 2% uranyl acetate for 7 min, and with Reynolds' lead citrate for 2 min. TEM images were captured using JEM-1011 (JEOL, Japan).

**Data availability**. The data supporting the findings of this study are available from the corresponding author upon reasonable request.

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

## Acknowledgements

We thank Y. Oh for his helpful discussions. This work was supported by two DGIST R&D Programs of the Ministry of Science, ICT&Future Planning (17-BD-06 and 17-01-HRMA-02) and Basic Science Research Program through the National Research Foundation of Korea (NRF) funded by the Ministry of Education (NRF-2016R1A6A3A11930198).

## Author contributions

J.Y.K. developed the AP-nanoPALDI MS system and performed experiments and data analysis. E.S.S. provided neuroscience expertise, prepared biological tissue specimens, and performed CARS measurements. H.K. set up the laser and optical path and devised gold nanorods experiments. J.-W.P. provided mass identification expertise and performed MS data analysis. D.-K.L. provided nanotechnology expertise and fabricated/provided surface-modified gold nanorods. D.W.M. designed and supervised the project. The manuscript was written by J.Y.K. and D.W.M. according to the discussion with all authors.

## Additional information

**Competing interests:** The authors declare no competing financial interests.

