## [Peer Review File · Nature Communications]

Reviewers' comments:

Reviewer #1 (Remarks to the Author):

This is an interesting manuscript that describes the development and performance of an exciting new instrument for ambient mass spectrometry imaging at high spatial resolutions. It represents a significant new advance in ambient mass spectrometry imaging of biological samples. One of the main challenges for ambient mass spectrometry imaging of biologic samples is the ability to acquire images at high spatial resolution. Techniques such as DESI typically produce images with spatial resolutions in the tens of microns range. MALDI can produce images approaching the micron range, but typically they are highly pixelated or the mass resolution is compromised. The instrument reported in this manuscript overcomes those limitations and produces images with a spatial resolution closer to ToF-SIMS. However, there are some concerns with the manuscript as written that the authors must address before it is publishable. These concerns are listed below.

I recommend publication after the authors have addressed the points listed below.

(1) Throughout the manuscript and even in the title the authors state they are imaging “live” tissues. This is not the case. Per the description in their experimental methods sections the animals are sacrificed and then the tissue sections are prepared for analysis. If the animals are sacrificed then the tissues they image can’t be considered to be “live.” I believe the authors meant to use the word “fresh” instead of “live.”

(2) The authors confuse pixel resolution with spatial resolution. Unfortunately this is rather common in the MALDI and atmospheric mass spec communities. They are acquiring their images in the microprobe mode, which means if they have sufficient pixel density the spatial resolution will be defined by the laser beam probe size. Although the authors state what their laser beam probe size is they don’t provide any actual documentation of the beam size. This can easily be done by running the probe across a straight edge. The authors must provide documentation of their probe size. This information can be included the supplementary information.

(3) The authors emphasize the spatial (x-y) resolution of their method, which is excellent for ambient mass spectrometry imaging. However, as they document in the manuscript the signal they use is integrated over a depth of ~35 microns. While collecting material over such a large

depth will certainly enhance their signal, it could compromise the spatial resolution for some tissues. For the examples shown in this manuscript their features likely are fairly constant in the x-y position over the 35 micron depth. However, this won't be the case for other tissue samples. For example, in tissue sections containing tumors the x-y distribution of the tumor can change significantly over a 35 micron depth. Thus, by collecting signal over a 35 micron depth the spatial resolution of the method will be compromised for those type of tissues. The authors need to address this point in the manuscript.

(4) The time to acquire the images shown in the manuscript must be specified. While the analyzer used by the authors to acquire the mass spectra provides excellent mass resolution and mass accuracy, the acquisition times are significantly slower than other analyzers (e.g., ToF analyzer). So it is important to specify the image acquisition time.

(5) Although the authors state in the experimental section that their mass analyzer is capable of providing mass resolutions of 35,000, they don't provide any experimental data to show they are actually achieve this mass resolution. Since this manuscript describes a new instrument it is important that the authors provide experimental evidence to document their achievable mass resolution. This can be added to the supplemental materials.

(6) The authors state the AuNRs were functionalization with mPEG-SH. They need to specify the chain length (MW or number of monomer units) of the PEG they used.

(7) Although the manuscript is reasonably well-written it would benefit from some light copy editing to improve the readability, as

Reviewer #2 (Remarks to the Author):

The paper describes a novel mass spectrometric tissue analysis technique using nanoparticle-assisted laser ablation coupled with atmospheric pressure chemical ionization using RF plasma as a source for primary reactant species. While the idea of this experimental setup is innovative, the presented experimental data does not meet the original expectations. The setup requires further refinement before it could be reported in a prestigious journal.

Although the authors present their technique as an ambient MS method, this statement is incorrect, since ambient MS assumes the analysis of unmodified samples under ambient conditions. Addition of matrix or nanoparticles deems the method to be non-ambient. This is not a criticism on the method but on the terminology used and the topics covered in the introduction part of the manuscript.

The comparison of the method with other techniques such as DESI or AP-MALDI is unfair in a number of cases. DESI has been reported to be destructive, being able to deliver in-depth

analytical capability, while AP-MALDI can easily be used for the analysis of tissue sections at the spatial resolution reported by the authors.

One of the most serious flaws of the paper is associated with the definition of spatial resolution. In MSI literature the spatial resolution is generally defined as feature resolution, so the diameter of the smallest anatomical or cellular feature detected by the method. In this sense the current study is expected to detect individual cells, as the 1.4 μm spatial resolution is enough to cover a single cell by tens of individual pixels. In contrast, individual cells are not visualized and the smallest detected features have 5-10 μm diameter although this is not reported directly by the authors.

The other fundamental problem with the study is the chemical information content of the information obtained. The authors report a very few ionic species without showing a single full, annotated spectrum (the two spectra in the Supplementary are not adequate for this purpose). The reported ions are actually not molecular ions but $[\text{M}+\text{H}-\text{H}_2\text{O}]^+$ in case of the MAGs and one cholesterol-associated species and $[\text{M}-\text{H}]^+$ for the other cholesterol-associated species. As far as the reviewer can tell all the identification is based on accurate mass measurements, which is not acceptable, especially not when only fragments of the expected species are detected.

Nevertheless, even if we accept the annotations, the conclusion is that the method can only detect a very few (7 reported) thermal degradation products of tissue components. State-of-the-art MALDI and SIMS methods can detect intact molecular ions of hundreds to thousands of tissue constituents at similar or better spatial resolution, making the use of the reported method hard to justify.

The choice of low temperature plasma for ionization is likely to be responsible for the poor analytical sensitivity and indirectly also for the poor spatial resolution. The LTP source also dries out the tissue slices, making the in-vivo claims of the authors highly questionable. The authors should consider using electrospray for post ionization (cf. LAESI) or use UV or mid-IR (2.94 μm) laser (lasers showing less scattering and more absorption by tissue) without post ionization in a similar setup.

In conclusion I suggest the rejection of the manuscript.

Reviewer #3 (Remarks to the Author):

General comment:

The manuscript presents a novel IMS technique named nanoPALDI, which relies on fine-tuned gold nanorods for desorption and a He plasma for ionization. While this new technique presents some interesting capabilities in terms of spatial resolution, it falls quite badly when it comes to the amount of detected analytes compared to other IMS methods capable of subcellular detection such as SIMS and MALDI. The analytical robustness of the presented MS data is quite poor and there is a dire need to properly identify the presented signals by tandem MS. Without formal

identification, all the biological interpretation of the data not supported. Although this new approach to IMS is quite interesting, this manuscript still needs some important work before being accepted for publication.

Major comments:

- Throughout the manuscript the authors identify several masses as monoacylglycerols (MAG), cholesterol, adenine or ceramide without ever mentioning how these identifications were performed. There is no information whether these were identified by exact mass followed by database search (and which database was used) or by MS/MS compared to a standard. The authors also fail to provide the ionization pathway these molecules are following. Are these detected analyte protonated or sodiated species? Did they undergo water loss as it would be the case for m/z 369.349 which is most likely [Cholesterol+H+-H₂O]? Exact mass is completely insufficient for identification when using complex samples without pre cleaning procedures. The only true confirmation is an overlapping of an experimental MS/MS from a standard with the MS/MS from the sample or comparing the experimental MS/MS with previously published material. The authors must provide a clear and detailed procedure on how these m/z were identified. If these were only identified by exact mass, please provide additional MS/MS spectra for (at least one) MAG, cholesterol, adenine and the ceramide fragment from tissue sections and compare these to the MS/MS of standards or at the very least from MS/MS found in the literature.

As a follow up, I would like to know how mass 385.346 could be cholesterol in the positive ionization mode when the exact mass of neutral cholesterol is 386.355. Do the authors truly believe cholesterol loses an H⁻ during ionization?

- I'm very surprised to review an imaging MS manuscript without any actual mass spectrum characteristic of the presented experiment. Could the authors provide an average spectrum of several "pixels" along with a single "pixel" mass spectrum from the hippocampus tissue analysis? This would help quite a bit in assessing the quality of the results.

- Page 10

I'm very surprised to see how many laser shots are being fired per "pixel". While the laser energy is quite low, can the authors comment on whether or not the tissue remains free of burn mark/damage around the ablation crater? Along with this, could the authors comment on why there is so few observed MS signals compared to what can be observed at such resolutions with other IMS method such as SIMS? Here again, presenting a mass spectrum would greatly help assess the overall number of detected molecules.

- Page 12

How homogeneous is mPEG-AuNR uptake in the tissue sections? Any uptake inhomogeneity's possibly due to differences in cell types or histologies will affect desorption yields and therefore

affect IMS results. This is a critical point. Please comment.

- Page 12 along with pages 31-32

Tissues sections are incubated in ACSF for 1h. Could the authors comment on the possible delocalization of certain analytes possibly occurring during this step? Could this explain why there is so few detected signals? Are all the hydrophilic compounds found in the section removed by this incubation step? I am more familiar with MALDI sample preparation and know for a fact that submersion of tissue sections in solvents (water or organic based) will inevitably delocalize certain analytes, i.e hydrophilic molecules in water based solutions (such as ACSF) and hydrophobic molecules in organic based solutions.

- Page 13 last paragraphs

Considering the steps and length of time it takes to prepare live tissue samples, how can the authors claim that nanoPALDI has little sample preparation compared to SIMS? In SIMS you only have to desiccate the tissue section after cryosectioning, which means waiting 10-20 min compared to the hour long incubation necessary for nanoPALDI. The authors should retract this statement.

Conclusion

“... without antibody tagging...”. To my knowledge, there is no available antibodies for the listed metabolites. So this sentence makes no sense.

Minor comments:

Page 7 – line 4

Change “desorption” to “ablation”

Page 10 – line 14

Change “bombarded” to “irradiated”

Page 12 – line 15

Strictly speaking, reference 40 is related to MALDI per se but to GALDI.

Figure 2

Please indicate the spatial resolution used in the figure caption.

Figure 3f

Please state what the error bars represent along with providing p values to asses statistical significance. This could simply be added to the supplementary Table 1 along with the raw data.

Figure 2, 3d and 5a

Is there a reason why there is no observable signal at m/z 369.349 in both figures 2 and 3 compared to the faint signal found in figure 5a?

Figure 5

Why are the two hippocampal regions a) and b) so different? Are they serial sections from the same animal undergoing different treatments or are they from two different animals? What types of cuts are we looking at (coronal, sagittal or horizontal)? At what depth (bregma) were the cuts made? This would help quite a bit with understanding hippocampus orientation and it would greatly facilitate the interpretation of the IMS data.

Answers to Reviewers' Comments

Title: *“Ambient Mass Spectrometric Imaging of Live Hippocampal Tissues with Subcellular Spatial Resolution”*

Manuscript ID: *NCOMMS-17-00257-T*

We found the Reviewers' comments very helpful in refining the manuscript. Below are our point-by-point responses to the Reviewers' comments.

Reviewer: 1

Recommendation: publication after the authors have addressed the points listed below.

I. Referee # 1 comments

This is an interesting manuscript that describes the development and performance of an exciting new instrument for ambient mass spectrometry imaging at high spatial resolutions. It represents a significant new advance in ambient mass spectrometry imaging of biological samples. One of the main challenges for ambient mass spectrometry imaging of biologic samples is the ability to acquire images at high spatial resolution. Techniques such as DESI typically produce images with spatial resolutions in the tens of microns range. MALDI can produce images approaching the micron range, but typically they are highly pixelated or the mass resolution is compromised. The instrument reported in this manuscript overcomes those limitations and produces images with a spatial resolution closer to ToF-SIMS. However, there are some concerns with the manuscript as written that the authors must address before it is publishable. These concerns are listed below.

I recommend publication after the authors have addressed the points listed below.

- We greatly appreciate your careful review of our manuscript and your positive comments and suggestions. We believe the manuscript has become much clearer and stronger as a result of addressing your comments.

First of all, note that the title has changed in the new manuscript. One of the reviewers suggested that “ambient MS imaging” should be changed to “atmospheric pressure MS imaging” because the tissue employed in this study were modified by AuNRs. We have decided to accept this suggestion, changing the title to “**Atmospheric Pressure Mass Spectrometry Imaging of Live Hippocampal Tissue Slices with**

Subcellular Spatial Resolution” to clarify the content and scope of the study. Accordingly, the term “ambient nano-PALDI” has been changed to “AP-nanoPALDI” throughout the manuscript. The Abstract, Introduction, and Conclusion have also been revised to describe atmospheric pressure (AP) MS imaging instead of ambient MS imaging.

(1) Throughout the manuscript and even in the title the authors state they are imaging “live” tissues. This is not the case. Per the description in their experimental methods sections the animals are sacrificed and then the tissue sections are prepared for analysis. If the animals are sacrificed then the tissues they image can’t be considered to be “live.” I believe the authors meant to use the word “fresh” instead of “live.”

- The term “live tissue slices” in reference to short-term living tissue slices is well-established in pseudo-two-dimensional models for research into neurophysiology, pathophysiology, and electrophysiology, in particular for studies in stretch-activated ion channels (SAC), microelectrode arrays (MEAs) of brain, cardiac, lung, and liver tissues (see references 33–35). Even if the mice were sacrificed for sample preparation in this study, the extracted hippocampal tissue slice was certainly viable until MS analysis began and could be cultured over several weeks. Therefore, the authors believe the resultant MS images are properly described as “live” tissue.

Nevertheless, based on the reviewer’s comment, we changed “live hippocampal tissues” to “live hippocampal tissue slices” in the title and the body to avoid ambiguity. Based on the reviewer’s comment, the sentences at lines 20–21 on page 6 and line 1–6 on page 7 and the related references are added in the new manuscript as follows:

“The term “live tissue slices” in reference to “short-term living tissue slices” is well-established in pseudo-two-dimensional models for research into neurophysiology, pathophysiology, and electrophysiology, in particular for studies in stretch-activated ion channels (SAC), microelectrode arrays (MEAs) of various organotypic tissues.³³⁻³⁵ In addition, even after treatment of AuNRs into the live hippocampal tissue slices, they still remained viable until desorption and ionization procedures. The detailed preparation of live hippocampal tissue slices and treatment of AuNRs for this study are described in the Method section.

33. Wang, K. *et al.* Living cardiac tissue slices: an organotypic pseudo two-dimensional model for cardiac biophysics research. *Prog. Biophys. Mol. Biol.* **115**, 314–327 (2014).

34. Opitz-Araya, X. & Barria, A. Organotypic hippocampal slice cultures. *J. Vis. Exp.* **48**, 2462 (2011).

35. Buskila, Y. *et al.* Extending the viability of acute brain slices. *Sci. Rep.* **4**, 5309 (2014).

The sentences at lines 9–12 on page 6 are added in the new manuscript as follows.

“Even though the tissue slice was treated by AuNRs, uptake of AuNRs is a natural biological behavior mainly caused by endocytosis of live tissues. Thus, tissue slices with AuNRs remained viable until MS analysis began.”

The word “live” was changed to “fresh” in several sentences to improve sentence fluency as follows.

At lines 9–10 on page 4:

“such as accurate molecular histology for live tissues and tissue-based drug screening.”
→ “such as accurate molecular histology for fresh tissues and tissue-based drug screening.”

At lines 9–10 on page 17:

“In addition, the proposed ambient nanoPALDI method can analyze the live tissue in the atmospheric pressure condition.” → “In addition, the proposed AP-nanoPALDI method can analyze the fresh tissue slice with humid in open-air AP and ambient temperature conditions.”

At lines 17–18 on page 17:

“Cholesterol images in **Fig. 3** and **Fig. 4** taken from live tissues were ~ ” → “Cholesterol images in **Fig. 3** and **Fig. 4** taken from fresh tissue slices were”

(2) The authors confuse pixel resolution with spatial resolution. Unfortunately this is rather common in the MALDI and atmospheric mass spec communities. They are acquiring their images in the microprobe mode, which means if they have sufficient pixel density the spatial resolution will be defined by the laser beam probe size. Although the authors state what their laser beam probe size is they don't provide any actual documentation of the beam size. This can easily be done by running the probe across a straight edge. The authors must provide documentation of their probe size. This information can be included the supplementary information.

- We agree with the reviewer's comment regarding the necessity of measurement of spatial resolution. In order to measure the spatial resolution (in particular, an x-axial

lateral resolution) of MS imaging, we ran the probe combined with the laser beam and non-thermal plasma jet across a straight edge of the specimen based on the reviewer's comment. To do so, the resolution test specimen was designed using a 400 rectangular mesh grid as a mask for preventing laser ablation. The hippocampal tissue slice treated with mPEG-AuNRs was placed onto the 400 mesh grid on the glass slide and MS analysis conducted. The resultant MS imaging from this specimen presented distinct border patterns, and the x-axial lateral resolutions were obtained to $2.9 \mu\text{m} \pm 0.6$ and $5.1 \mu\text{m} \pm 1.1$ when the sample moving velocities were $10 \mu\text{m/s}$ and $30 \mu\text{m/s}$, respectively.

As a result of the measurement, the measured resolution and calculated pixel size are found to be entirely different in definition and in actual length. Thus, pixel size and spatial resolution are carefully distinguished to avoid ambiguity in the new manuscript. In addition, the phrase at line 1 on page 18, "while keeping the subcellular spatial resolution almost the same as SIMS" changes to "while keeping the subcellular spatial resolution" in the new manuscript. The detailed experimental descriptions and the related figure and references about the spatial resolution of MS imaging are added in the new manuscript as follows.

At lines 3–21 on page 11 and lines 1–3 on page 12:

"To investigate the spatial resolution of the AP-nanoPALDI method, the resolution test specimen was designed using a gold TEM grid with 400 mesh so that the resultant MS imaging would have several straight edges, as shown in **Fig. 5**. A hippocampal tissue slice treated with mPEG-AuNRs was placed onto a 400 rectangular mesh grid (G400, GilderGrids, UK) on the glass slide and MS analyses were conducted. As the gold mesh grid played the role of a mask for preventing laser ablation, the resultant MS imaging presented distinct border patterns, as shown in **Fig. 5**. The 400 rectangular mesh grid employed in this test had a pitch of $62 \mu\text{m}$, mesh openings of $37 \times 37 \mu\text{m}^2$ separated by $25\text{-}\mu\text{m}$ bars, and a $20\text{-}\mu\text{m}$ thickness. **Fig. 5a,b** shows the ion images of the MAG (16:0) at $m/z = 313.273$ from the hippocampal tissue slice when the sample moving velocities for raster scanning were set differently to $10 \mu\text{m/s}$ and $30 \mu\text{m/s}$, respectively. The x-axial lateral resolution of the MAG (16:0) ion image has been derived from the average of the three neighboring scan lines, as shown in **Fig. 5c,d**. The distance across which the signal change from 16 to 84% (or 84 to 16%) of the maximum indicated an x-axial lateral resolution.^{41,42} As a result, x-axial lateral resolutions of these ion images of $2.9 \mu\text{m} \pm 0.6$ and $5.1 \mu\text{m} \pm 1.1$ were achieved (means \pm SDs from 10 distinct edges are shown in both cases of **Fig. 5 c,d**.) when the sample moving velocities were $10 \mu\text{m/s}$ and $30 \mu\text{m/s}$, respectively. In summary, the ion images in **Fig. 3** had an x-axial pixel

size of 4.2 μm and an x-axial lateral resolution of 5.1 μm , and the ion images in Fig. 4 had an x-axial pixel size of 1.4 μm and an x-axial resolution of 2.9 μm according to the change of sample moving velocity. In both cases of Fig. 3 and Fig. 4, the y-axial pixel size and resolution are equal to 5 μm , which was determined by the spacing of the scan lines.”

41. Saka, S. K. *et al.* Correlated optical and isotopic nanoscopy. *Nat. Commun.* **5**, 3664 (2014).
42. Ghosal, S. *et al.* Imaging and 3D elemental characterization of intact bacterial spores by high-resolution secondary ion mass spectrometry. *Anal. Chem.* **80**, 5986–5992 (2008).

The authors add Fig. 5 in the new manuscript as follows.

Figure 5. The x-axial lateral resolution test of AP-nanoPALDI imaging in hippocampal specimen on 400 mesh grid.

The sentence “Therefore, we think that the estimated spatial resolution of 1.4 μm is not arbitrary but real due to the use of AuNRs and ultrafast laser in the ambient air condition.” in the old manuscript has also been eliminated.

In **Supplementary Note 2**, the pixel size and spatial resolution are distinguished to avoid ambiguity, and sentences at lines 16–18 on page 3 in Supplementary Information are added as follows.

“Besides, as mentioned in the Results section of the main text, the x-axial lateral resolutions of these MS images were 2.9 μm and 5.1 μm when the sample moving velocities were 10 $\mu\text{m/s}$ and 30 $\mu\text{m/s}$ (Fig. 5c,d), respectively.”

(3) The authors emphasize the spatial (x-y) resolution of their method, which is excellent for ambient mass spectrometry imaging. However, as they document in the manuscript the signal they use is integrated over a depth of ~ 35 microns. While collecting material over such a large depth will certainly enhance their signal, it could compromise the spatial resolution for some tissues. For the examples shown in this manuscript their features likely are fairly constant in the x-y position over the 35 micron depth. However, this won't be the case for other tissue samples. For example, in tissue sections containing tumors the x-y distribution of the tumor can change significantly over a 35 micron depth. Thus, by collecting signal over a 35 micron depth the spatial resolution of the method will be compromised for those type of tissues. The authors need to address this point in the manuscript.

- As a matter of fact, since short-term living tissue slices are primarily used as pseudo-two-dimensional models, these MS images do not consider depth resolution or depth information. Rather, we thought that the integrated signal over 35 μm in the depth direction could improve the reliability of the experimental results. However, I understand the reviewer's concern over poor depth resolution. According to the reviewer's comment, a description about sampling depth for MS analysis has been added in the new manuscript as follows.

At lines 20–22 on page 15 and lines 1–3 on page 16:

“It was noticed that the high aspect ratio of the laser-ablated crater was detrimental to the spatial resolution in the depth direction. The tissue slices, pseudo-two-dimensional organotypic models, could be normally analyzed with the AP-nanoPALDI method without difficulty. However, because of a limitation of the depth resolution in the AP-nanoPALDI method, careful analysis and interpretation were required when analyzing inhomogeneous tissue specimens in the depth direction, such as cancer tissues.”

(4) The time to acquire the images shown in the manuscript must be specified. While the analyzer used by the authors to acquire the mass spectra provides excellent mass resolution and mass accuracy, the acquisition times are significantly slower than other analyzers (e.g., ToF analyzer). So it is important to specify the image acquisition time.

- In accordance with the reviewer's comment, the total acquisition times for MS

imaging are specified in the new manuscript. The acquisition time of the MS data was 70 seconds per scan line for all MS images in **Fig. 3**, **Fig. 4**, and **Fig. 7**, comprising 60 seconds to measure one x-axial scan line and a 10 second pause for preparing measurement of the next line. Thus, total data acquisition times for one MS imaging were approximately two hours (117 min.) for 100 lines and slightly less than 6 hours for 300 lines (350 min) in this study. The description about the acquisition time for MS images has been added in the new manuscript as follows.

At lines 11–14 on page 9:

“Data acquisition took 70 seconds per scan line: 60 seconds to measure one x-axial scan line and a 10-second pause for preparing a measurement of the next line. Thus, the total data acquisition time for the MS imaging was 350 minutes (70 seconds per line scan × 300 lines)”

At lines 12–13 on page 10:

“~ and total data acquisition time was 117 min (70 seconds per line scan × 100 lines).”

(5) Although the authors state in the experimental section that their mass analyzer is capable of providing mass resolutions of 35,000, they don't provide any experimental data to show they are actually achieve this mass resolution. Since this manuscript describes a new instrument it is important that the authors provide experimental evidence to document their achievable mass resolution. This can be added to the supplemental materials.

- The Q-Exactive hybrid quadrupole-Orbitrap (QE Orbitrap) mass analyzer employed in this study can support high resolving power, up to 140,000 FWHM at $m/z = 200$. However, with a higher mass resolution setting, the internal analysis time (scan rate) of the analyzer is also inevitably increased, resulting in large pixel size of MS imaging (See **Supplementary Note 2**). Therefore, the mass resolution was set to a lower level (35,000) for MS imaging. This resolving power is not an excessive parametric value at all in the QE Orbitrap mass analyzer.

According to the reviewer's comment, the actual mass spectra, their descriptions and the table summarizing the mass identification are added in the new manuscript as follows.

At lines 7–22 on page 7, lines 1-22 on page 8, and line 1 on page 9:

“The ions detected by AP-nanoPALDI method from the live hippocampal tissue slice of an adult Institute of Cancer Research (ICR) mouse are presented in **Fig. 2a**. Because the

plasma is an ionized gas, many ions were detected from the plasma itself; thus, the plasma background has been excluded from spectra. The MS spectra, as shown in **Fig. 2a**, represented the average of 400 consecutive scans for data reliability, and all spectra were obtained in positive ion mode. A positive ion mass spectrum recorded from nonthermal AP plasma is shown in **Supplementary Fig. 5**. Despite the mass range being set to $m/z = 100$ to 1,000 for measurements, strong ion signals were mostly observed under $m/z = 500$ from mouse hippocampal tissues. An examination of the MS spectra indicated the presence of more than 200 specimen-related ion species, and some of the detected ions were assigned to particular lipids and metabolites. A Q-Exactive hybrid quadrupole-Orbitrap (QE Orbitrap, Thermo Fisher Scientific, Germany) mass spectrometer was used to measure the exact mass of each peak in full scan mode, and each chemical formula was assigned using the XCalibur 3.0 software. With a strict comparison of the measured masses and the calculated chemical formulas, mass spectrum peaks observed in the mouse hippocampal tissue slices were identified as metabolites, lipids, and their derivatives, such as adenine, cholesterol, phosphocholine, and several fragments of glycerolipids and sphingolipids, as shown in **Table 1**. In addition, the assignments of cholesterol ($m/z = 385.346$) and adenine ($m/z = 136.061$), peaks were confirmed by a comparison of tandem MS data with the corresponding standard materials. The AP-nanoPALDI MS/MS spectra of the ion at $m/z = 385.346$ from the hippocampal tissue slices showed identical fragmentation patterns to a standard of cholesterol, as shown in **Fig. 2b,c**. In the same manner, the AP-nanoPALDI MS/MS spectra of the ion at $m/z = 136.061$ also showed a fragmentation pattern identical to those of a standard of adenine (**Fig. 3d,e**). Unsaturated and saturated MAG ions ($m/z = 311.258$, $m/z = 313.273$, $m/z = 337.273$, $m/z = 339.289$, and $m/z = 341.305$) were reported to be well detectable in brain tissues in the positive ion mode³⁶⁻³⁸ and were found to be precisely in agreement with chemical formulas, as shown in **Table 1**. In addition, by careful comparisons among the tandem MS data of the MAG ions, it was found that saturated MAG ions have distinctive fragmentation patterns while unsaturated MAG ions do not exhibit their distinctive fragmentation patterns (see **Supplementary Fig. 6**). Several biomolecules including fragments of sphingolipids were further assigned. Many of the ions assigned to $[M+H-H_2O]^+$ were in the form of H_2O subtracted from the base ions $[M+H]^+$, which seems to be related to the experimental environment, such as humid specimen and open-air AP condition. It is important to note that signal intensities of the ions assigned to $[M+H-H_2O]^+$ proved sufficiently strong to construct MS imaging in general. In particular, only the cholesterol ion is seen as $[M-H]^+$ at $m/z = 385.346$ and $[M+H-H_2O]^+$ at $m/z = 369.352$.

It is commonly reported that the spectra showed main ions at $m/z = 369.4$ and $m/z = 385.4$, respectively corresponding to $[M+H-H_2O]^+$ and $[M-H]^+$ for cholesterol in SIMS and laser ablation using MS experiments.^{37,39,40,,}

The related References are also added as follows:

36. Seyer, A. *et al.* Lipidomic and spatio-temporal imaging of fat by mass spectrometry in mice duodenum during lipid digestion. *PLoS ONE*. **8**, e58224 (2013).
37. Vaikkinen, A. *et al.* Infrared laser ablation atmospheric pressure photoionization mass spectrometry. *Anal. Chem.* **84**, 1630–1636 (2012).
38. Ikeda, K. *et al.* Global analysis of triacylglycerols including oxidized molecular species by reverse-phase high resolution LC/ESI-QTOF MS/MS. *J. Chromatogr. B Analyt. Technol. Biomed. Life Sci.* **877**, 2639–2647 (2009).
39. Piehowski, P. D. *et al.* MS/MS methodology to improve subcellular mapping of cholesterol using TOF-SIMS. *Anal. Chem.* **80**, 8662–8667 (2008).
40. Altelaar, A. F. *et al.* Gold-enhanced biomolecular surface imaging of cells and tissue by SIMS and MALDI mass spectrometry. *Anal. Chem.* **78**, 734–742 (2006).

Fig. 2 and Table 1 have also been added as follows:

Figure 2. AP-nanoPALDI mass spectra from a mouse hippocampal tissue slice and standard materials in the positive ion mode.

Table 1. Assigned lipids, metabolites, and derivatives from mouse hippocampal tissue slices in the positive ion mode using AP-nanoPALDI MS.

Compound		Measured m/z	Theoretical m/z	Error (p pm)	Molecular fo rmula	Species	Ref.
Sterol lipid	Cholesterol	385.3459	385.3464	1.30	$C_{27}H_{45}O$	$[M-H]^-$	37
		369.3521	369.3515	-1.62	$C_{27}H_{45}$	$[M+H-H_2O]^-$	39
Glycerolipid	MAG 16:1	311.2575	311.2581	1.93	$C_{19}H_{35}O_3$	$[M+H-H_2O]^-$	36
		329.2692	329.2686	-1.82	$C_{19}H_{37}O_4$	$[M+H]^-$	69
	MAG 16:0	313.2731	313.2737	1.92	$C_{19}H_{37}O_3$	$[M+H-H_2O]^-$	36
		331.2851	331.2843	-2.41	$C_{19}H_{39}O_4$	$[M+H]^-$	69
	MAG 18:2	337.2732	337.2737	1.48	$C_{21}H_{37}O_3$	$[M+H-H_2O]^-$	36
		355.2851	355.2843	-2.25	$C_{21}H_{39}O_4$	$[M+H]^-$	69
	MAG 18:1	339.2889	339.2894	1.47	$C_{21}H_{39}O_3$	$[M+H-H_2O]^-$	36
		357.3007	357.2999	-2.24	$C_{21}H_{41}O_4$	$[M+H]^-$	69
	MAG 18:0	341.3046	341.3050	1.17	$C_{21}H_{41}O_3$	$[M+H-H_2O]^-$	36
		359.3161	359.3156	-1.39	$C_{21}H_{43}O_4$	$[[M+H]^-$	69
Sphingolipid	Ceramide 18:0	548.5390	548.5401	2.01	$C_{36}H_{70}NO_2$	$[M+H-H_2O]^-$	70
		282.2785	282.2791	2.13	$C_{18}H_{36}NO$	$[M+H-H_2O]^-$	71
	Sphingosine	300.2905	300.2897	-2.66	$C_{18}H_{38}NO_2$	$[M+H]^-$	71
		284.2945	284.2947	0.70	$C_{18}H_{38}NO$	$[M+H-H_2O]^-$	71
	Sphinganine	302.3062	302.3054	-2.65	$C_{18}H_{40}NO_2$	$[M+H]^-$	71
Adenine		136.0614	136.0617	2.20	$C_8H_6N_6$	$[M+H]^-$	53
Phosphocholine		184.0754	184.0733	-11.41	$C_8H_{14}NO_2P$	$[M]^-$	40

Supplementary Fig. 5,6 are added in the Supplementary Information.

Supplementary Fig. 5 A positive ion mass spectrum recorded from nonthermal atmospheric pressure helium plasma using AP-nanoPALDI.

Supplementary Fig. 6 AP-nanoPALDI MS/MS data of saturated MAG ions from the mouse hippocampal tissue slice.

In addition, the experimental parameters for tandem MS analysis are added in the mass analyzer part of the Method section as follows.

At lines 6–11 on page 32:

“Parameters for the tandem MS scan were as follows: mass resolution of 17,500 FWHM, isolation window of 0.4 m/z, high-energy collisional dissociation (HCD) with normalized collision energy (NCE) of 20 – 80, AGC target of 2.0×10^5 , maximum injection time of 100 ms, underfill ratio of 1.0 %, exclude isotopes “on”, and dynamic exclusion 5.0 s. Using Inclusion and Exclusion Indexes, the particular ion peaks were analyzed with MS/MS.”

(6) The authors state the AuNRs were functionalized with mPEG-SH. They need to specify the chain length (MW or number of monomer units) of the PEG they used.

- The AuNRs were functionalized with mPEG-SH (MW = 6000 Da). In accordance with the reviewer's comment, the sentence at lines 11–12 on page 35 is revised in the new manuscript as follows:

“Into this solution, mPEG-SH (MW 6,000, 2.0 mg) was added, followed by the addition of 10 mL ethanol.”

(7) Although the manuscript is reasonably well-written it would benefit from some light copy editing to improve the readability, as

- According to the reviewer's comment, the manuscript has been corrected with a full revision. And redundancies have been eliminated. I really appreciate the reviewer's comments.

Note: Ion images of MAG (18:2), sphingosine, and sphinganine were added; the descriptions about ion images have been revised and intensified as follows.

At lines 10–11 on page 2:

“~ such as fragments of glycerolipid and sphingolipid, adenine, and cholesterol.”

At lines 15–19 on page 9:

“~ions at $m/z = 311.258$ of monoacylglycerol (MAG) (16:1), $m/z = 313.273$ of MAG (16:0), $m/z = 337.273$ of MAG (18:2), $m/z = 339.289$ of MAG (18:1), $m/z = 341.305$ of MAG (18:0), $m/z = 136.061$ of adenine, $m/z = 282.279$ of sphingosine, $m/z = 284.295$ of sphinganine (18:0), $m/z = 385.346$ of cholesterol, and $m/z = 548.539$ of ceramide (18:0).”

At lines 16–22 on page 19:

“~ monoacylglycerols, sphingosine, sphinganine, cholesterol, and ceramide ions compared in detail (**Fig. 4e,f** and **Supplementary Table 1**). The signal intensities of MAG ions (at $m/z = 311.258$, $m/z = 313.273$, $m/z = 337.273$, $m/z = 339.289$, and $m/z = 341.305$) in the basal dendrite side were 48–72% of those in the apical dendrite side. In contrast, the ceramide ion at $m/z = 548.539$ was observed to be similar on both sides. The high-resolution MS imaging from the AP-nanoPALDI method clearly revealed that MAGs are distributed 1.4–2.1 times more ~”

At lines 4–7 on page 20:

“With these high-resolution MS images, we found that the apical dendrites contained more monoacylglycerols, sphingosine, sphinganine, and cholesterol than the basal dendrites, while the ceramide seems to have even distributions in the two dendritic structures in the CA3 of the live hippocampus tissue.”

At lines 17–19 on page 22:

“Using AP-nanoPALDI, clear MS images of bio-molecules, including fragments of sphingolipid and glycerophospholipid, adenine, and cholesterol, were obtained from mouse hippocampal tissue slices without staining procedures.”

II. Referee # 2

Recommendation: rejection of the manuscript

The paper describes a novel mass spectrometric tissue analysis technique using nanoparticle-assisted laser ablation coupled with atmospheric pressure chemical ionization using RF plasma as a source for primary reactant species. While the idea of this experimental setup is innovative, the presented experimental data does not meet the original expectations.

- We greatly appreciate your careful and insightful comments. We believe the manuscript has become much clearer and stronger as a result of addressing them.

The setup requires further refinement before it could be reported in a prestigious journal. Although the authors present their technique as an ambient MS method, this statement is incorrect, since ambient MS assumes the analysis of unmodified samples under ambient conditions. Addition of matrix or nanoparticles deems the method to be non-ambient. This is not a criticism on the method but on the terminology used and the topics covered in the introduction part of the manuscript.

- We agree with the reviewer's comment and concern about a use of the term "Ambient nanoPALDI" to refer to an ambient MS method. In this study, the desorption property of the hippocampal tissue by lasers was clearly improved by AuNRs treatment so that the tissue employed in this study were not in a native state, as noted by the reviewer. The sample desorption and ionization in the proposed MS method proceeded under open-air atmospheric pressure and ambient temperature (no additional heating) conditions. Moreover, the tissue slice with mPEG-AuNR remained viable until desorption and ionization procedures, because the uptake of AuNRs is a natural biological behavior mainly caused by endocytosis of live tissues. Strictly speaking, however, this proposed MS method is an atmospheric pressure MS method but not an ambient MS method, as pointed out by the reviewer. Accordingly, first, the title was changed to "**Atmospheric Pressure Mass Spectrometry Imaging of Live Hippocampal Tissue Slices with Subcellular Spatial Resolution**" to clarify the content and scope of this study. Second, the term "ambient nano-PALDI" has been changed to "AP-nanoPALDI" throughout the manuscript. Lastly, the Abstract, Introduction, and Conclusion sections have also been revised to describe the AP-nanoPALDI as a new of atmospheric pressure MS imaging method as follows.

At lines 1–3 on page 2:

"We report a high spatial resolution mass spectrometry (MS) system that allows us to sensitively image live hippocampal tissue slices under open-air atmospheric pressure

(AP) and ambient temperature conditions at the subcellular level.”

At lines 2–9 on page 3:

“Recently, atmospheric pressure (AP) ionization mass spectrometry, which can directly analyze samples with minimal or no sample preparation, has fascinated researchers in fields other than analytical chemistry because it seems to have potential applications in various research fields.¹⁻¹⁰ Applications have been demonstrated for pesticide or explosive detections and acquisition of molecular information in biological samples using various MS methods at AP.¹¹⁻¹³ In addition, AP mass spectrometry (AP-MS) imaging, which can provide both chemical and spatial information has also been studied with a number of different AP-ionization methods.”

The related references are also changed.

1. Seitzinger, S. P. *et al.* Atmospheric pressure mass spectrometry: a new analytical chemical characterization method for dissolved organic matter in rainwater. *Environ. Sci. Technol.* **37**, 131–137 (2003).
2. Li, D. X. *et al.* Gas chromatography coupled to atmospheric pressure ionization mass spectrometry (GC-API-MS): review. *Anal. Chim. Acta* **891**, 43–61 (2015).
3. Bruins, A. P. Atmospheric-pressure-ionization mass spectrometry: I. Instrumentation and ionization techniques. *TrAC, Trends Anal. Chem.* **13**, Pages 37–43 (1994).
4. Chen, H. & Zenobi, R. Neutral desorption sampling of biological surfaces for rapid chemical characterization by extractive electrospray ionization mass spectrometry. *Nat. Protoc.* **3**, 1467–1475 (2008).
5. Suni, N. M. *et al.* Analysis of lipids with desorption atmospheric pressure photoionization-mass spectrometry (DAPPI-MS) and desorption electrospray ionization-mass spectrometry (DESI-MS). *J. Mass Spectrom.* **47**, 611–619 (2012).
6. Saf, R. *et al.* Thin organic films by atmospheric-pressure ion deposition. *Nat. Mater.* **3**, 323–329 (2004).
7. Raffaelli, A. & Saba, A. Atmospheric pressure photoionization mass spectrometry. *Mass Spectrom. Rev.* **22**, 318–331 (2003).
10. Cooks, R. *et al.* Ambient mass spectrometry. *Science* **311**, 1566–1570 (2006).

The comparison of the method with other techniques such as DESI or AP-MALDI is unfair in a number of cases. DESI has been reported to be destructive, being able to deliver in-depth analytical capability, while AP-MALDI can easily be used for the analysis of tissue sections at the spatial resolution reported by the authors.

- We appreciate your comment, which improved the quality of our manuscript. We emphasize in the Introduction section that AP-nanoPALDI method is capable of analyzing living tissues with high spatial resolution at atmospheric pressure. Regarding the comparison of the proposed AP-nanoPALDI with DESI and AP-MALDI, the disadvantages of DESI and AP-MALDI are not critical in the introduction, so the following sentences, “AP-MALDI cannot be applied for live cells and tissues.¹⁵ DESI can be applied to live tissues but is only sensitive to the surface and near surface due to its intrinsic mechanism of sampling. It depends on the dissolved molecular constituents of the surface that have become encapsulated in ejected secondary droplets.^{1,14}” have been eliminated.

In addition, a sentence is revised at line 20 on page 3 in the new manuscript as follows:

“In order for AP-MS to be more widely used for biomedical research, ~”

One of the most serious flaws of the paper is associated with the definition of spatial resolution. In MSI literature the spatial resolution is generally defined as feature resolution, so the diameter of the smallest anatomical or cellular feature detected by the method. In this sense the current study is expected to detect individual cells, as the 1.4 μm spatial resolution is enough to cover a single cell by tens of individual pixels. In contrast, individual cells are not visualized and the smallest detected features have 5-10 μm diameter although this is not reported directly by the authors.

- We agree with the reviewer’s comment concerning the necessity of measurement of spatial resolution. In order to measure the spatial resolution (in particular, x-axial lateral resolution) of MS imaging, we ran the probe combined with the laser beam and non-thermal plasma jet across a straight edge of the specimen. To do so, the resolution test specimen was designed using a 400 rectangular mesh grid as a mask for preventing laser ablation. The hippocampal tissue slice treated with mPEG-AuNRs was placed onto the 400 mesh grid on the glass slide and subjected to MS analysis. The resultant MS imaging from this specimen presented distinct border patterns, and the x-axial lateral resolutions obtained are $2.9 \mu\text{m} \pm 0.6$ and $5.1 \mu\text{m} \pm 1.1$ when the sample moving velocities were $10 \mu\text{m/s}$ and $30 \mu\text{m/s}$, respectively. As a result of the spatial resolution, the measured resolution and calculated pixel size are entirely different in definition and in actual length. Thus, pixel size and spatial resolution have been clearly distinguished in the new manuscript to avoid ambiguity.

Even if the spatial resolution achieves subcellular level, since the hippocampal tissue consists of numerous number of arbor-structured neuron cells densely connected to each other and establishing a substantial neural networking, individual cells cannot be

effectively visualized in hippocampal tissue slices with MS imaging. Instead, we can observe the different molecular compositions between apical and basal dendrites, as depicted in **Fig. 4**. Because a single neuron cell structurally divides to soma, apical dendrite, and basal dendrite, the high spatial resolution MS image can provide the molecular information of the subcellular region of neuron cells in the form of ensemble average. In particular, the location of the soma was clearly defined by the spatial distribution of adenine ions.

The detailed experimental descriptions and the related figure and references about the spatial resolution of MS imaging are added in the new manuscript as follows.

At lines 3–21 on page 11 and lines 1–3 on page 12:

“To investigate the spatial resolution of the AP-nanoPALDI method, the resolution test specimen was designed using a gold TEM grid with 400 mesh so that the resultant MS imaging would have several straight edges, as shown in **Fig. 5**. A hippocampal tissue slice treated with mPEG-AuNRs was placed onto a 400 rectangular mesh grid (G400, GilderGrids, UK) on the glass slide and MS analyses were conducted. As the gold mesh grid played the role of a mask for preventing laser ablation, the resultant MS imaging presented distinct border patterns, as shown in **Fig. 5**. The 400 rectangular mesh grid employed in this test had a pitch of 62 μm , mesh openings of $37 \times 37 \mu\text{m}^2$ separated by 25- μm bars, and a 20- μm thickness. **Fig. 5a,b** shows the ion images of the MAG (16:0) at $m/z = 313.273$ from the hippocampal tissue slice when the sample moving velocities for raster scanning were set differently to 10 $\mu\text{m/s}$ and 30 $\mu\text{m/s}$, respectively. The x-axial lateral resolution of the MAG (16:0) ion image has been derived from the average of the three neighboring scan lines, as shown in **Fig. 5c,d**. The distance across which the signal change from 16 to 84% (or 84 to 16%) of the maximum indicated an x-axial lateral resolution.^{41,42} As a result, x-axial lateral resolutions of these ion images of $2.9 \mu\text{m} \pm 0.6$ and $5.1 \mu\text{m} \pm 1.1$ were achieved (means \pm SDs from 10 distinct edges are shown in both cases of **Fig. 5 c,d**.) when the sample moving velocities were 10 $\mu\text{m/s}$ and 30 $\mu\text{m/s}$, respectively. In summary, the ion images in **Fig. 3** had an x-axial pixel size of 4.2 μm and an x-axial lateral resolution of 5.1 μm , and the ion images in **Fig. 4** had an x-axial pixel size of 1.4 μm and an x-axial resolution of 2.9 μm according to the change of sample moving velocity. In both cases of **Fig. 3** and **Fig. 4**, the y-axial pixel size and resolution are equal to 5 μm , which was determined by the spacing of the scan lines.”

41. Saka, S. K. *et al.* Correlated optical and isotopic nanoscopy. *Nat. Commun.* **5**, 3664

(2014).

42. Ghosal, S. *et al.* Imaging and 3D elemental characterization of intact bacterial spores by high-resolution secondary ion mass spectrometry. *Anal. Chem.* **80**, 5986–5992 (2008).

The authors add **Fig. 5** in the new manuscript as follows.

Figure 5. The x-axial lateral resolution test of AP-nanoPALDI imaging in hippocampal specimen on 400 mesh grid.

The sentence “Therefore, we think that the estimated spatial resolution of 1.4 μm is not arbitrary but real due to the use of AuNRs and ultrafast laser in the ambient air condition.” has also been eliminated.

In **Supplementary Note 2**, the pixel size and spatial resolution are distinguished to avoid ambiguity, and sentences at lines 16–19 on page 3 in Supplementary Information are added as follows.

“Besides, as mentioned in the Result section of the main body, the x-axial lateral resolutions of these MS images were achieved to 2.9 μm and 5.1 μm when the sample moving velocities were 10 $\mu\text{m}/\text{s}$ and 30 $\mu\text{m}/\text{s}$ (**Fig. 5c,d**), respectively.”

The other fundamental problem with the study is the chemical information content of the information obtained. The authors report a very few ionic species without showing a single full, annotated spectrum (the two spectra in the Supplementary are not adequate for this purpose). The reported ions are actually not molecular ions but $[M+H-H_2O]^+$ in case of the MAGs and one cholesterol-associated species and $[M-H]^+$ for the other cholesterol-associated species. As far as the reviewer can tell all the identification is based on accurate mass measurements, which is not acceptable, especially not when only fragments of the expected species are detected. Nevertheless, even if we accept the annotations, the conclusion is that the method can only detect a very few (7 reported) thermal degradation products of tissue components. State-of-the-art MALDI and SIMS methods can detect intact molecular ions of hundreds to thousands of tissue constituents at similar or better spatial resolution, making the use of the reported method hard to justify.

- We understand the reviewer's concern over the lack of chemical information content including mass identification. In accordance with the reviewer's comment, the actual mass spectra, their descriptions, and the table summarizing the mass identification are added in the new manuscript as shown below. As shown in **Table 1**, the measured and theoretical m/z values are shown to be quite consistent due to the high-resolution mass analyzer (QE Orbitrap). As shown in the MS spectra, over 200 specimen-related ion images were observed in the range of $m/z = 100$ to 1000 from the live hippocampal tissue slice of the adult mouse. The authors showed only eight ion images in the old manuscript; based on reviewer's comment, we add more assigned molecules and organize the list including chemical information as shown in **Table 1**. Ion images of MAG (18:2), sphingosine, and sphinganine have been added in **Fig. 3** and **Fig. 4**, and ion images of sphingosine and sphinganine have been added in **Fig. 7**.

The descriptions and the table summarizing the mass identification are added in the new manuscript as follows.

At lines 7–22 on page 7, lines 1–22 on page 8, and line 1 on page 9:

“The ions detected by AP-nanoPALDI method from the live hippocampal tissue slice of an adult Institute of Cancer Research (ICR) mouse are presented in **Fig. 2a**. Because the plasma is an ionized gas, many ions were detected from the plasma itself; thus, the plasma background has been excluded from spectra. The MS spectra, as shown in **Fig. 2a**, represented the average of 400 consecutive scans for data reliability, and all spectra were obtained in positive ion mode. A positive ion mass spectrum recorded from nonthermal AP plasma is shown in **Supplementary Fig. 5**. Despite the mass range being set to $m/z = 100$ to 1,000 for measurements, strong ion signals were mostly observed under $m/z = 500$ from mouse hippocampal tissues. An examination of the MS spectra indicated the presence of more than 200 specimen-related ion species, and some

of the detected ions were assigned to particular lipids and metabolites. A Q-Exactive hybrid quadrupole-Orbitrap (QE Orbitrap, Thermo Fisher Scientific, Germany) mass spectrometer was used to measure the exact mass of each peak in full scan mode, and each chemical formula was assigned using the XCalibur 3.0 software. With a strict comparison of the measured masses and the calculated chemical formulas, mass spectrum peaks observed in the mouse hippocampal tissue slices were identified as metabolites, lipids, and their derivatives, such as adenine, cholesterol, phosphocholine, and several fragments of glycerolipids and sphingolipids, as shown in **Table 1**. In addition, the assignments of cholesterol ($m/z = 385.346$) and adenine ($m/z = 136.061$), peaks were confirmed by a comparison of tandem MS data with the corresponding standard materials. The AP-nanoPALDI MS/MS spectra of the ion at $m/z = 385.346$ from the hippocampal tissue slices showed identical fragmentation patterns to a standard of cholesterol, as shown in **Fig. 2b,c**. In the same manner, the AP-nanoPALDI MS/MS spectra of the ion at $m/z = 136.061$ also showed a fragmentation pattern identical to those of a standard of adenine (**Fig. 3d,e**). Unsaturated and saturated MAG ions ($m/z = 311.258$, $m/z = 313.273$, $m/z = 337.273$, $m/z = 339.289$, and $m/z = 341.305$) were reported to be well detectable in brain tissues in the positive ion mode³⁶⁻³⁸ and were found to be precisely in agreement with chemical formulas, as shown in **Table 1**. In addition, by careful comparisons among the tandem MS data of the MAG ions, it was found that saturated MAG ions have distinctive fragmentation patterns while unsaturated MAG ions do not exhibit their distinctive fragmentation patterns (see **Supplementary Fig. 6**). Several biomolecules including fragments of sphingolipids were further assigned. Many of the ions assigned to $[M+H-H_2O]^+$ were in the form of H_2O subtracted from the base ions $[M+H]^+$, which seems to be related to the experimental environment, such as humid specimen and open-air AP condition. It is important to note that signal intensities of the ions assigned to $[M+H-H_2O]^+$ proved sufficiently strong to construct MS imaging in general. In particular, only the cholesterol ion is seen as $[M-H]^+$ at $m/z = 385.346$ and $[M+H-H_2O]^+$ at $m/z = 369.352$. It is commonly reported that the spectra showed main ions at $m/z = 369.4$ and $m/z = 385.4$, respectively corresponding to $[M+H-H_2O]^+$ and $[M-H]^+$ for cholesterol in SIMS and laser ablation using MS experiments.^{37,39,40,,}

The related References are also added as follows:

36. Seyer, A. *et al.* Lipidomic and spatio-temporal imaging of fat by mass spectrometry in mice duodenum during lipid digestion. *PLoS ONE*. **8**, e58224 (2013).
37. Vaikkinen, A. *et al.* Infrared laser ablation atmospheric pressure photoionization

- mass spectrometry. *Anal. Chem.* **84**, 1630–1636 (2012).
38. Ikeda, K. *et al.* Global analysis of triacylglycerols including oxidized molecular species by reverse-phase high resolution LC/ESI-QTOF MS/MS. *J. Chromatogr. B Analyt. Technol. Biomed. Life Sci.* **877**, 2639–2647 (2009).
39. Piehowski, P. D. *et al.* MS/MS methodology to improve subcellular mapping of cholesterol using TOF-SIMS. *Anal. Chem.* **80**, 8662–8667 (2008).
40. Altelaar, A. F. *et al.* Gold-enhanced biomolecular surface imaging of cells and tissue by SIMS and MALDI mass spectrometry. *Anal. Chem.* **78**, 734–742 (2006).

Fig. 2 and Table 1 have also been added as follows.

Figure 2. AP-nanoPALDI mass spectra from a mouse hippocampal tissue slice and standard materials in the positive ion mode.

Table 1. Assigned lipids, metabolites, and derivatives from mouse hippocampal tissue slices in the positive ion mode using AP-nanoPALDI MS.

Compound		Measured m/z	Theoretical m/z	Error (p pm)	Molecular fo rmula	Species	Ref.
Sterol lipid	Cholesterol	385.3459	385.3464	1.30	C ₂₇ H ₄₅ O	[M-H] ⁻	37
		369.3521	369.3515	-1.62	C ₂₇ H ₄₅	[M+H-H ₂ O] ⁻	39
Glycerolipid	MAG 16:1	311.2575	311.2581	1.93	C ₁₉ H ₃₅ O ₃	[M+H-H ₂ O] ⁻	36
		329.2692	329.2686	-1.82	C ₁₉ H ₃₇ O ₄	[M+H] ⁻	69
	MAG 16:0	313.2731	313.2737	1.92	C ₁₉ H ₃₇ O ₃	[M+H-H ₂ O] ⁻	36
		331.2851	331.2843	-2.41	C ₁₉ H ₃₉ O ₄	[M+H] ⁻	69
	MAG 18:2	337.2732	337.2737	1.48	C ₂₁ H ₃₇ O ₃	[M+H-H ₂ O] ⁻	36
		355.2851	355.2843	-2.25	C ₂₁ H ₃₉ O ₄	[M+H] ⁻	69
	MAG 18:1	339.2889	339.2894	1.47	C ₂₁ H ₃₉ O ₃	[M+H-H ₂ O] ⁻	36
		357.3007	357.2999	-2.24	C ₂₁ H ₄₁ O ₄	[M+H] ⁻	69
	MAG 18:0	341.3046	341.3050	1.17	C ₂₁ H ₄₁ O ₃	[M+H-H ₂ O] ⁻	36
		359.3161	359.3156	-1.39	C ₂₁ H ₄₃ O ₄	[[M+H] ⁻	69
Sphingolipid	Ceramide 18:0	548.5390	548.5401	2.01	C ₃₆ H ₇₀ NO ₂	[M+H-H ₂ O] ⁻	70
		282.2785	282.2791	2.13	C ₁₈ H ₃₆ NO	[M+H-H ₂ O] ⁻	71
	300.2905	300.2897	-2.66	C ₁₈ H ₃₈ NO ₂	[M+H] ⁻	71	
	Sphinganine	284.2945	284.2947	0.70	C ₁₈ H ₃₈ NO	[M+H-H ₂ O] ⁻	71
		302.3062	302.3054	-2.65	C ₁₈ H ₄₀ NO ₂	[M+H] ⁻	71
Adenine		136.0614	136.0617	2.20	C ₂ H ₄ N ₅	[M+H] ⁻	53
Phosphocholine		184.0754	184.0733	-11.41	C ₂ H ₁₄ NO ₂ P	[M] ⁻	40

Supplementary Fig. 5,6 are added in the Supplementary Information.

Supplementary Fig. 5 A positive ion mass spectrum recorded from nonthermal atmospheric pressure helium plasma using AP-nanoPALDI.

Supplementary Fig. 6 AP-nanoPALDI MS/MS data of saturated MAG ions from the mouse hippocampal tissue slice.

In addition, the experimental parameters for tandem MS analysis are added in the mass analyzer part of the Method section as follows.

At lines 6–11 on page 32:

“Parameters for the tandem MS scan were as follows: mass resolution of 17,500 FWHM, isolation window of 0.4 m/z, high-energy collisional dissociation (HCD) with normalized collision energy (NCE) of 20 – 80, AGC target of 2.0×10^5 , maximum injection time of 100 ms, underfill ratio of 1.0 %, exclude isotopes “on”, and dynamic exclusion 5.0 s. Using Inclusion and Exclusion Indexes, the particular ion peaks were analyzed with MS/MS.”

Note: Ion images of MAG (18:2), sphingosine, and sphinganine were added; the descriptions about ion images have been revised and intensified as follows.

At lines 10–11 on page 2:

“~ such as fragments of glycerolipid and sphingolipid, adenine, and cholesterol.”

At lines 15–19 on page 9:

“~ions at $m/z = 311.258$ of monoacylglycerol (MAG) (16:1), $m/z = 313.273$ of MAG (16:0), $m/z = 337.273$ of MAG (18:2), $m/z = 339.289$ of MAG (18:1), $m/z = 341.305$ of MAG (18:0), $m/z = 136.061$ of adenine, $m/z = 282.279$ of sphingosine, $m/z = 284.295$ of sphinganine (18:0), $m/z = 385.346$ of cholesterol, and $m/z = 548.539$ of ceramide (18:0).”

At lines 16–22 on page 19:

“~ monoacylglycerols, sphingosine, sphinganine, cholesterol, and ceramide ions compared in detail (**Fig. 4e,f** and **Supplementary Table 1**). The signal intensities of MAG ions (at $m/z = 311.258$, $m/z = 313.273$, $m/z = 337.273$, $m/z = 339.289$, and $m/z = 341.305$) in the basal dendrite side were 48–72% of those in the apical dendrite side. In contrast, the ceramide ion at $m/z = 548.539$ was observed to be similar on both sides. The high-resolution MS imaging from the AP-nanoPALDI method clearly revealed that MAGs are distributed 1.4–2.1 times more ~”

At lines 4–7 on page 20:

“With these high-resolution MS images, we found that the apical dendrites contained more monoacylglycerols, sphingosine, sphinganine, and cholesterol than the basal dendrites, while the ceramide seems to have even distributions in the two dendritic structures in the CA3 of the live hippocampus tissue.”

At lines 17–19 on page 22:

“Using AP-nanoPALDI, clear MS images of bio-molecules, including fragments of sphingolipid and glycerophospholipid, adenine, and cholesterol, were obtained from mouse hippocampal tissue slices without staining procedures.”

The choice of low temperature plasma for ionization is likely to be responsible for the poor analytical sensitivity and indirectly also for the poor spatial resolution. The LTP source also dries out the tissue slices, making the in-vivo claims of the authors highly questionable. The authors should consider using electrospray for post ionization (cf. LAESI) or use UV or mid-IR (2.94 μm) laser (lasers showing less scattering and more absorption by tissue) without post ionization in a similar setup.

In conclusion I suggest the rejection of the manuscript.

- We believe that the spatial resolution of our MS imaging is not poor compared to any other ambient IMS method. Moreover, we do not think that the nonthermal plasma jet in our method dries out the tissue slices. The plasma plume does not contact the specimen directly due to the additional pumping by gas-flow-assisted ion transport equipment as shown in **Supplementary Fig. 2c,d**.

Nonetheless, in accordance with the reviewer's comment, we recognize analytical sensitivity problems used by a nonthermal atmospheric pressure plasma jet for a post-ionization method; low signal intensity in the mass range of ions of $m/z > 500$ in the positive ion mode and non-availability in negative ion mode for tissue analysis. Therefore, we already consider the use of electrospray for post ionization instead of the nonthermal plasma jet device. However, to install an electrospray into this MS system, we need to re-design and install the ionization source part and the stage where specimen desorption/ionization takes place; this work is beyond the main purpose of this paper, introducing the high resolution MS imaging of live tissue slice at atmospheric pressure. The modified method with an electrospray will be reported later through further experiments. We greatly appreciate your insightful comment.

III. Referee # 3 comments

Recommendation: needs some important work before being accepted for publication

General comment:

The manuscript presents a novel IMS technique named nanoPALDI, which relies on fine-tuned gold nanorods for desorption and a He plasma for ionization. While this new technique presents some interesting capabilities in terms of spatial resolution, it falls quite badly when it comes to the amount of detected analytes compared to other IMS methods capable of subcellular detection such as SIMS and MALDI. The analytical robustness of the presented MS data is quite poor and there is a dire need to properly identify the presented signals by tandem MS.

Without formal identification, all the biological interpretation of the data not supported. Although this new approach to IMS is quite interesting, this manuscript still needs some important work before being accepted for publication.

- We greatly appreciate your careful review of our manuscript and insightful comments and concerns. We believe the manuscript has become much clearer and stronger as a result of addressing them.

First of all, note that the title has changed in the new manuscript. One of the reviewers suggested that “ambient MS imaging” should be changed to “atmospheric pressure MS imaging” because the tissues employed in this study were modified by AuNRs. We have decided to accept this suggestion, and the title has been revised to “**Atmospheric Pressure Mass Spectrometry Imaging of Live Hippocampal Tissue Slices with Subcellular Spatial Resolution**” to clarify the content and scope of the study. Accordingly, the term “ambient nano-PALDI” has been changed to “AP-nanoPALDI” throughout the manuscript. The Abstract, Introduction, and Conclusion are also revised to describe atmospheric pressure (AP) MS imaging instead of ambient MS imaging.

Major comments:

- Throughout the manuscript the authors identify several masses as monoacylglycerols (MAG), cholesterol, adenine or ceramide without ever mentioning how these identifications were performed. There is no information whether these were identified by exact mass followed by database search (and which database was used) or by MS/MS compared to a standard. The authors also fail to provide the ionization pathway these molecules are following. Are these detected analyte protonated or sodiated species? Did they undergo water lost as it would be the case for m/z 369.349 which is most likely [Cholesterol+H⁺-H₂O]? Exact mass is completely insufficient for identification when using complexes samples without pre-cleaning procedures. The only true confirmation is an overlapping of an experimental MS/MS from a standard with the MS/MS from the sample or comparing the experimental MS/MS with previously published material. The authors must provide a clear and detailed procedure

on how these m/z were identified. If these were only identified by exact mass, please provide additional MS/MS spectra for (at least one) MAG, cholesterol, adenine and the ceramide fragment from tissue sections and compare these to the MS/MS of standards or at the very least from MS/MS found in the literature. As a follow up, I would like to know how mass 385.346 could be cholesterol in the positive ionization mode when the exact mass of neutral cholesterol is 386.355. Do the authors truly believe cholesterol loses an H- during ionization?

- We appreciate the reviewer's careful review and valuable comments, which certainly improved the quality of our manuscript. We agree with the reviewer's comment concerning the absence of any mass identification procedure. Based on the reviewer's comment, the mass identification including chemical information of the detected ions is described in detail in the new manuscript as shown below. The authors have also added a table (**Table 1**) summarizing the mass identification in the new manuscript. The main assigned ions, $[M+H-H_2O]^+$, were in the form of H₂O subtracted from the base ions, $[M+H]^+$, which seems to be related to the experimental environment, such as humid specimen and open-air atmospheric pressure condition. In particular, only the cholesterol ion is seen as $[M+H-H_2O]^+$ at m/z = 369.352 and $[M-H]^+$ at m/z = 385.346. It is commonly reported that the spectra showed main ions at m/z = 369.4 and m/z = 385.4 corresponding to $[M+H-H_2O]^+$ and $[M-H]^+$ for cholesterol in SIMS and laser ablation using MS experiments.

(Note: Regarding the cholesterol peak at m/z = 369.349 specified in the old manuscript, the ion peak has been modified from m/z = 369.349 to m/z = 369.352 in the new manuscript by careful and accurate spectral monitoring.)

At lines 7–22 on page 7, lines 1–22 on page 8, and line 1 on page 9:

“The ions detected by AP-nanoPALDI method from the live hippocampal tissue slice of an adult Institute of Cancer Research (ICR) mouse are presented in **Fig. 2a**. Because the plasma is an ionized gas, many ions were detected from the plasma itself; thus, the plasma background has been excluded from spectra. The MS spectra, as shown in **Fig. 2a**, represented the average of 400 consecutive scans for data reliability, and all spectra were obtained in positive ion mode. A positive ion mass spectrum recorded from nonthermal AP plasma is shown in **Supplementary Fig. 5**. Despite the mass range being set to m/z = 100 to 1,000 for measurements, strong ion signals were mostly observed under m/z = 500 from mouse hippocampal tissues. An examination of the MS spectra indicated the presence of more than 200 specimen-related ion species, and some of the detected ions were assigned to particular lipids and metabolites. A Q-Exactive hybrid quadrupole-Orbitrap (QE Orbitrap, Thermo Fisher Scientific, Germany) mass spectrometer was used to measure the exact mass of each peak in full scan mode, and

each chemical formula was assigned using the XCalibur 3.0 software. With a strict comparison of the measured masses and the calculated chemical formulas, mass spectrum peaks observed in the mouse hippocampal tissue slices were identified as metabolites, lipids, and their derivatives, such as adenine, cholesterol, phosphocholine, and several fragments of glycerolipids and sphingolipids, as shown in **Table 1**. In addition, the assignments of cholesterol ($m/z = 385.346$) and adenine ($m/z = 136.061$), peaks were confirmed by a comparison of tandem MS data with the corresponding standard materials. The AP-nanoPALDI MS/MS spectra of the ion at $m/z = 385.346$ from the hippocampal tissue slices showed identical fragmentation patterns to a standard of cholesterol, as shown in **Fig. 2b,c**. In the same manner, the AP-nanoPALDI MS/MS spectra of the ion at $m/z = 136.061$ also showed a fragmentation pattern identical to those of a standard of adenine (**Fig. 3d,e**). Unsaturated and saturated MAG ions ($m/z = 311.258$, $m/z = 313.273$, $m/z = 337.273$, $m/z = 339.289$, and $m/z = 341.305$) were reported to be well detectable in brain tissues in the positive ion mode³⁶⁻³⁸ and were found to be precisely in agreement with chemical formulas, as shown in **Table 1**. In addition, by careful comparisons among the tandem MS data of the MAG ions, it was found that saturated MAG ions have distinctive fragmentation patterns while unsaturated MAG ions do not exhibit their distinctive fragmentation patterns (see **Supplementary Fig. 6**). Several biomolecules including fragments of sphingolipids were further assigned. Many of the ions assigned to $[M+H-H_2O]^+$ were in the form of H_2O subtracted from the base ions $[M+H]^+$, which seems to be related to the experimental environment, such as humid specimen and open-air AP condition. It is important to note that signal intensities of the ions assigned to $[M+H-H_2O]^+$ proved sufficiently strong to construct MS imaging in general. In particular, only the cholesterol ion is seen as $[M-H]^+$ at $m/z = 385.346$ and $[M+H-H_2O]^+$ at $m/z = 369.352$. It is commonly reported that the spectra showed main ions at $m/z = 369.4$ and $m/z = 385.4$, respectively corresponding to $[M+H-H_2O]^+$ and $[M-H]^+$ for cholesterol in SIMS and laser ablation using MS experiments.^{37,39,40}

The related References are also added as follows:

36. Seyer, A. *et al.* Lipidomic and spatio-temporal imaging of fat by mass spectrometry in mice duodenum during lipid digestion. *PLoS ONE*. **8**, e58224 (2013).
37. Vaikkinen, A. *et al.* Infrared laser ablation atmospheric pressure photoionization mass spectrometry. *Anal. Chem.* **84**, 1630–1636 (2012).
38. Ikeda, K. *et al.* Global analysis of triacylglycerols including oxidized molecular species by reverse-phase high resolution LC/ESI-QTOF MS/MS. *J. Chromatogr. B*

Analyt. Technol. Biomed. Life Sci. **877**, 2639–2647 (2009).

39. Piehowski, P. D. *et al.* MS/MS methodology to improve subcellular mapping of cholesterol using TOF-SIMS. *Anal. Chem.* **80**, 8662–8667 (2008).

40. Altaar, A. F. *et al.* Gold-enhanced biomolecular surface imaging of cells and tissue by SIMS and MALDI mass spectrometry. *Anal. Chem.* **78**, 734–742 (2006).

Fig. 2 and **Table 1** have also been added as follows.

Figure 2. AP-nanoPALDI mass spectra from a mouse hippocampal tissue slice and standard materials in the positive ion mode.

Table 1. Assigned lipids, metabolites, and derivatives from mouse hippocampal tissue slices in the positive ion mode using AP-nanoPALDI MS.

Compound		Measured m/z	Theoretical m/z	Error (p pm)	Molecular fo rmula	Species	Ref.
Sterol lipid	Cholesterol	385.3459	385.3464	1.30	C ₂₇ H ₄₅ O	[M-H] ⁻	37
		369.3521	369.3515	-1.62	C ₂₇ H ₄₅	[M+H-H ₂ O] ⁻	39
Glycerolipid	MAG 16:1	311.2575	311.2581	1.93	C ₁₉ H ₃₅ O ₃	[M+H-H ₂ O] ⁻	36
		329.2692	329.2686	-1.82	C ₁₉ H ₃₇ O ₄	[M+H] ⁻	69
	MAG 16:0	313.2731	313.2737	1.92	C ₁₉ H ₃₇ O ₃	[M+H-H ₂ O] ⁻	36
		331.2851	331.2843	-2.41	C ₁₉ H ₃₉ O ₄	[M+H] ⁻	69
	MAG 18:2	337.2732	337.2737	1.48	C ₂₁ H ₃₇ O ₃	[M+H-H ₂ O] ⁻	36
		355.2851	355.2843	-2.25	C ₂₁ H ₃₉ O ₄	[M+H] ⁻	69
	MAG 18:1	339.2889	339.2894	1.47	C ₂₁ H ₃₉ O ₃	[M+H-H ₂ O] ⁻	36
		357.3007	357.2999	-2.24	C ₂₁ H ₄₁ O ₄	[M+H] ⁻	69
	MAG 18:0	341.3046	341.3050	1.17	C ₂₁ H ₄₁ O ₃	[M+H-H ₂ O] ⁻	36
		359.3161	359.3156	-1.39	C ₂₁ H ₄₃ O ₄	[[M+H] ⁻	69
Sphingolipid	Ceramide 18:0	548.5390	548.5401	2.01	C ₃₆ H ₇₀ NO ₂	[M+H-H ₂ O] ⁻	70
		282.2785	282.2791	2.13	C ₁₈ H ₃₆ NO	[M+H-H ₂ O] ⁻	71
	300.2905	300.2897	-2.66	C ₁₈ H ₃₈ NO ₂	[M+H] ⁻	71	
	Sphinganine	284.2945	284.2947	0.70	C ₁₈ H ₃₈ NO	[M+H-H ₂ O] ⁻	71
		302.3062	302.3054	-2.65	C ₁₈ H ₄₀ NO ₂	[M+H] ⁻	71
Adenine		136.0614	136.0617	2.20	C ₂ H ₄ N ₅	[M+H] ⁻	53
Phosphocholine		184.0754	184.0733	-11.41	C ₂ H ₁₄ NO ₂ P	[M] ⁻	40

Supplementary Fig. 5,6 are added in the Supplementary Information.

Supplementary Fig. 5 A positive ion mass spectrum recorded from nonthermal atmospheric pressure helium plasma using AP-nanoPALDI.

Supplementary Fig. 6 AP-nanoPALDI MS/MS data of saturated MAG ions from the mouse hippocampal tissue slice.

In addition, the experimental parameters for tandem MS analysis are added in the mass analyzer part of the Method section as follows.

At lines 6–11 on page 32:

“Parameters for the tandem MS scan were as follows: mass resolution of 17,500 FWHM, isolation window of 0.4 m/z, high-energy collisional dissociation (HCD) with normalized collision energy (NCE) of 20 – 80, AGC target of 2.0×10^5 , maximum injection time of 100 ms, underfill ratio of 1.0 %, exclude isotopes “on”, and dynamic exclusion 5.0 s. Using Inclusion and Exclusion Indexes, the particular ion peaks were analyzed with MS/MS.”

- I'm very surprised to review an imaging MS manuscript without any actual mass spectrum characteristic of the presented experiment. Could the authors provide an average spectrum of several "pixels" along with a single "pixel" mass spectrum from the hippocampus tissue analysis? This would help quite a bit in assessing the quality of the results.

- In accordance with the reviewer's comment, the actual ion spectra detected by AP-nanoPALDI method from the live hippocampal tissue slice of an adult mouse have been added in the new manuscript as depicted above.

The following figure (**Fig. R1**) showed (a) a single scan (pixel) mass spectrum and (b) an average spectrum of 400 consecutive scans (pixels). The plasma background has been subtracted from the spectra. Since one scan line of the MS image was slightly over 400 pixels (433 pixels) in this study, the 400 consecutive spectra were averaged for data reliability. As shown in **Fig. R1a,b**, the average spectrum showed that ion peaks with low signal intensity were suppressed, and those with high signal intensity were highlighted by the data averaging effect, compared to the single scan mass spectrum. The MS spectra as shown in **Fig. 2a** of the manuscript represented the average of 400 consecutive scans for data reliability.

Figure R1. Mass spectra for (a) a single scan (pixel) and (b) an average spectrum of 400 consecutive scans (pixels) from the hippocampus tissue analysis.

- Page 10

I'm very surprised to see how many laser shots are being fired per "pixel". While the laser energy is quite low, can the authors comment on whether or not the tissue remains free of burn mark/damage around the ablation crater? Along with this, could the authors comment on why there is so few observed MS signals compared to what can be observed at such resolutions with other IMS method such as SIMS? Here again, presenting a mass spectrum would greatly help assess the overall number of detected molecules.

- We can confirm that there is no significant obstruction of MS analysis due to burning damage. As shown in **Fig. 6g** in the new manuscript, the linear craters observed by helium ion microscopy are consistently quite steep and clear, and no damage caused by the burn around the linear crater was found. Even if the thermal desorption by laser ablation occurred in the tissue slice, since the area where the thermal desorption occurs instantaneously is very small compared to the entire analysis region of the specimen, no errors due to the burn have been found.

The authors understand the importance of showing the mass spectra and present mass spectra from the tissue slice in the new manuscript. Actually, over 200 specimen-related ion images were observed in the range of $m/z = 100$ to 1000. Based on the reviewer's comment, we have added more assigned molecules and organized the list including chemical information, as shown in **Table 1** in the new manuscript.

Based on the reviewer's comment, the related sentence has been added at lines 7–8 on page 17 in the new manuscript:

“Therefore, despite continuous irradiation with a large number of laser shots, no damage caused by the burn around the linear craters was found, as shown in **Fig. 6g**.”

- Page 12

How homogeneous is mPEG-AuNR uptake in the tissue sections? Any uptake inhomogeneity's possibly due to differences in cell types or histologies will affect desorption yields and therefore affect IMS results. This is a critical point. Please comment.

- In order to determine the distribution of AuNRs in the tissue, TEM images were obtained. They confirmed that AuNRs employed in this study were fairly evenly distributed throughout the tissue. Because the particle size of mPEG-AuNRs is very small compared to the size of the focused laser beam, it is not necessary that the nanoparticles are ideally evenly distributed in the tissue. The descriptions about the distribution of mPEG-AuNR throughout the tissue have been added in the new manuscript as follows, and the TEM images are presented in **Supplementary Fig. 7**.

At lines 5–12 on page 15:

“TEM images were obtained to investigate the actual distribution of AuNRs inside the tissue as shown in **Supplementary Fig 7**. TEM images confirmed that mPEG-AuNRs were fairly evenly distributed throughout the tissue. Because the particle size of mPEG-AuNRs was very small compared to the size of the focused laser beam, the nanoparticles were not necessary to be ideally evenly distributed in the tissue. However, it was also found that the desorption performance depended on the concentration of the AuNRs, thus the concentration of mPEG-AuNRs was experimentally optimized.”

At lines 3–14 on page 38:

Transmission electron microscopy (TEM) imaging Primary fixation was accomplished at 4°C for 4 h using 2% paraformaldehyde and 2% glutaraldehyde in 0.05 M sodium cacodylate buffer at pH 7.2. For post-fixation, 1% osmium tetroxide in 0.05 M sodium cacodylate buffer was used at 4 °C for 2 h. *En bloc* staining was performed at 4°C for 30 min with 0.5 % uranyl acetate. The samples were dehydrated at room temperature for 10 min at each step with 30%, 50%, 70%, 80%, 90%, 100%, 100%, and 100% ethanol. Transition was performed twice at room temperature for 15 min with 100% propylene oxide. Infiltration was performed using Spurr’s resin. Polymerization was performed at 70 °C for 24 h. The samples were sectioned to a thickness of 80–120 nm by Ultramicrotome (MT-X, RMC, USA), and then stained with 2% uranyl acetate for 7 min, and with Reynolds’ lead citrate for 2 min. TEM images were captured using JEM-1011 (JEOL, Japan).

Supplementary Fig. 7 is added as follows.

Supplementary Fig. 7 TEM images of hippocampal tissue slice treated with mPEG-AuNRs for AP-nanoPALDI MS, each using the same tissue slice with different magnifications.

- Page 12 along with pages 31-32

Tissues sections are incubated in ACSF for 1h. Could the authors comment on the possible delocalization of certain analytes possibly occurring during this step? Could this explain why there is so few detected signals? Are all the hydrophilic compounds found in the section removed by this incubation step? I am more familiar with MALDI sample preparation and know for a fact that submersion of tissue sections in solvents (water or organic based) will inevitably delocalize certain analytes, i.e hydrophilic molecules in water based solutions (such as ACSF) and hydrophobic molecules in organic based solutions.

- The tissue slice with a thickness of 200 μm employed in this study is sufficiently viable for quite a while. Thus, we can assure that there is no significant delocalization of analytes by solution media (ACSF) during aeration and incubation procedures because the preparation of the tissue slice follows a well-established method. Aeration is a necessary procedure to remove debris from tissue damaged by chopping and to secure membrane-terminated tissue slice and incubation is necessary for stabilization of the

tissue slices and the uptake of gold nanorods by endocytosis. We understand your concern about our sample preparation being different from SIMS or MALDI. If the thickness of the tissue slice is as thin as ten to twenty micrometers as with the sample preparation used in SIMS or MALDI, the viability of the tissue slices will be substantially low, then a problem of delocalization of analytes might occur, as you mentioned.

In fact, over 200 specimen-related ion species were observed in the range of $m/z = 100\text{--}1000$. The authors showed only eight ion images in the old manuscript. Based on the reviewer's comment, we have added more assigned molecules and organized the list including chemical information in **Table 1**. We appreciate the reviewer's careful comments, which certainly improved the quality of our manuscript.

Based on the reviewer's comment, the sentences at lines 20–21 on page 6 and line 1–6 on page 7 and the related references are added in the new manuscript as follows:

“The term “live tissue slices” in reference to “short-term living tissue slices” is well-established in pseudo-two-dimensional models for research into neurophysiology, pathophysiology, and electrophysiology, in particular for studies in stretch-activated ion channels (SAC), microelectrode arrays (MEAs) of various organotypic tissues.³³⁻³⁵ In addition, even after treatment of AuNRs into the live hippocampal tissue slices, they still remained viable until desorption and ionization procedures. The detailed preparation of live hippocampal tissue slices and treatment of AuNRs for this study are described in the Method section.

33. Wang, K. *et al.* Living cardiac tissue slices: an organotypic pseudo two-dimensional model for cardiac biophysics research. *Prog. Biophys. Mol. Biol.* **115**, 314–327 (2014).
34. Opitz-Araya, X. & Barria, A. Organotypic hippocampal slice cultures. *J. Vis. Exp.* **48**, 2462 (2011).
35. Buskila, Y. *et al.* Extending the viability of acute brain slices. *Sci. Rep.* **4**, 5309 (2014).

The sentences at lines 9–12 on page 6 are added in the new manuscript as follows.
“Even though the tissue slice was treated by AuNRs, uptake of AuNRs is a natural biological behavior mainly caused by endocytosis of live tissues. Thus, tissue slices with AuNRs remained viable until MS analysis began.”

- Page 13 last paragraphs
Considering the steps and length of time it takes to prepare live tissue samples, how can the authors claim that nanoPALDI has little sample preparation compared to SIMS? In SIMS you only have to desiccate the tissue section after cryosectioning, which means waiting 10-20 min compared to the hour long incubation necessary for nanoPALDI. The authors should retract this statement.

- We agree with the reviewer's comment. Accordingly, the sentence "Thus, the MS images achieved by the ambient nanoPALDI-MS have the advantage of little sample preparation compared with vacuum-based secondary ion mass spectrometry (SIMS), which has been effectively used for lipid imaging with high spatial resolution (~1 µm), especially for brain imaging." has been eliminated. We appreciate the reviewer's careful comment, which improved the quality of our manuscript.

Conclusion
"… without antibody tagging…". To my knowledge, there is no available antibodies for the listed metabolites. So this sentence makes no sense.

- We agree with the reviewer's comment. Accordingly, the phrase "~ antibody tagging or ~" has been eliminated.

Minor comments:
Page 7 – line 4
Change "desorption" to "ablation"

- The authors revised this in the new manuscript.

Page 10 – line 14
Change "bombarded" to "irradiated"

- The authors revised this in the new manuscript.

Page 12 – line 15
Strictly speaking, reference 40 is related to MALDI per se but to GALDI.

- The authors agree with your comment and have changed the reference to the SALDI-related journal, as follows, in the new manuscript. In addition, the authors also added and re-numbered the references as needed.

49. Lo, C.-Y. *et al.* Surface assisted laser desorption/ionization mass spectrometry on titania nanotube arrays. *J. Am. Soc. Mass Spectrom.* **19**, 1014–1020 (2008).

Figure 2
Please indicate the spatial resolution used in the figure caption.

- The authors have indicated the spatial resolution information in the figure captions of

Fig. 3 and **Fig. 4** in the new manuscript as follows.

In the figure caption of **Fig. 3** on page 42:

“~ and the corresponding x-axial and y-axial pixel sizes were 4.2 μm and 5 μm , respectively. These ion images had x-axial and y-axial lateral resolutions of 5.1 μm and 5 μm , respectively (**Fig. 5**).”

In the figure caption of **Fig. 4** on page 44:

“~ and the corresponding x-axial and y-axial pixel sizes were 1.4 μm and 5 μm , respectively. According to experiment, the ion images had x-axial and y-axial lateral resolutions of 2.9 μm and 5 μm , respectively (**Fig. 5**).”

Figure 3f

Please state what the error bars represent along with providing p values to assess statistical significance. This could simply be added to the supplementary Table 1 along with the raw data.

- Based on the reviewer's comment, **Fig. 4f** has been modified as follows. We have provided *P* values, and error bars that represented standard deviations. In addition, the sentences about the caption of **Fig. 4f** have been refined as follows.

At the figure caption of **Fig. 4f** on page 44:

“(f) Relative proportions of detected ions, where the intensities of apical dendritic populated areas were normalized to 100% (raw data is shown in **Supplementary Table 1**). Shown are means \pm SDs from nine independent ions. Asterisks indicated *P* values.

******, $P < 0.01$; *******, $P < 0.001$.”

We appreciate the reviewer’s careful comment and suggestion, which certainly improved the quality of our manuscript.

Figure 2, 3d and 5a

Is there a reason why there is no observable signal at m/z 369.349 in both figures 2 and 3 compared to the faint signal found in figure 5a?

- The difference in signal intensities at $m/z = 369.352$ in **Fig. 2**, **Fig. 3**, and **Fig 5a** of the old manuscript was caused by an individual difference or variation of the specimens. (Note: Accurate spectral monitoring led to the cholesterol peak being modified from $m/z = 369.349$ to $m/z = 369.352$ in the new manuscript.) The ion peak at $m/z = 369.352$ represented a cholesterol peak with H_2O subtracted from $m/z = 385.346$. Because the signal intensity at $m/z = 385.346$ is higher than $m/z = 369.352$, the ion image of $m/z = 385.346$ can clearly provide the spatial distribution of cholesterol in the tissue slice. Therefore, in order to avoid ambiguity, the authors decided to remove the ion images of $m/z = 369.352$ in **Fig. 3**, **Fig. 4**, and **Fig. 7** of the new manuscript after a serious discussion between the co-authors. Instead, ion images of MAG (18:2), sphingosine, and sphinganine have been added in **Fig. 3** and **Fig. 4**, and ion images of sphingosine and sphinganine are added in **Fig. 7**.

Figure 5

Why are the two hippocampal regions a) and b) so different? Are they serial sections from the same animal undergoing different treatments or are they from two different animals? What types of cuts are we looking at (coronal, sagittal or horizontal)? At what depth (bregma) were the cuts made? This would help quite a bit with understanding hippocampus orientation and it would greatly facilitate the interpretation of the IMS data.

- Two hippocampal tissue slices in **Fig. 7** of the new manuscript were serial sections from the same animal undergoing different treatments. According to the sample preparation protocol, after incubation for 1 h, the tissue slices were put on the slide glasses by pipetting. Therefore, the soft tissue slices on the slides could not be an identical shape in spite of serial sections from the same hippocampus. In addition, the analysis regions were also slightly different. Note that the tissue slices employed in this study were never frozen. The shapes of the prepared tissue slices of 200- μ m thickness cannot be the same, contrary to serial sections of cryo-sectioned tissues.

Based on the reviewer’s comment, the sentence at lines 20–21 on page 22 and line 1 on page 23 has been added in the new manuscript as follows:

“It was noteworthy that these two hippocampal tissue slices were serial sections from the same ICR mouse in spite of difference in shape, because the specimens were prepared without being frozen.”

Note: Ion images of MAG (18:2), sphingosine, and sphinganine were added; the descriptions about ion images have been revised and intensified as follows.

At lines 10–11 on page 2:

“~ such as fragments of glycerolipid and sphingolipid, adenine, and cholesterol.”

At lines 15–19 on page 9:

“~ions at $m/z = 311.258$ of monoacylglycerol (MAG) (16:1), $m/z = 313.273$ of MAG (16:0), $m/z = 337.273$ of MAG (18:2), $m/z = 339.289$ of MAG (18:1), $m/z = 341.305$ of MAG (18:0), $m/z = 136.061$ of adenine, $m/z = 282.279$ of sphingosine, $m/z = 284.295$ of sphinganine (18:0), $m/z = 385.346$ of cholesterol, and $m/z = 548.539$ of ceramide (18:0).”

At lines 16–22 on page 19:

“~ monoacylglycerols, sphingosine, sphinganine, cholesterol, and ceramide ions compared in detail (**Fig. 4e,f** and **Supplementary Table 1**). The signal intensities of MAG ions (at $m/z = 311.258$, $m/z = 313.273$, $m/z = 337.273$, $m/z = 339.289$, and $m/z = 341.305$) in the basal dendrite side were 48–72% of those in the apical dendrite side. In contrast, the ceramide ion at $m/z = 548.539$ was observed to be similar on both sides. The high-resolution MS imaging from the AP-nanoPALDI method clearly revealed that MAGs are distributed 1.4–2.1 times more ~”

At lines 4–7 on page 20:

“With these high-resolution MS images, we found that the apical dendrites contained more monoacylglycerols, sphingosine, sphinganine, and cholesterol than the basal dendrites, while the ceramide seems to have even distributions in the two dendritic structures in the CA3 of the live hippocampus tissue.”

At lines 17–19 on page 22:

“Using AP-nanoPALDI, clear MS images of bio-molecules, including fragments of sphingolipid and glycerophospholipid, adenine, and cholesterol, were obtained from mouse hippocampal tissue slices without staining procedures.”

Reviewers' comments:

Reviewer #1 (Remarks to the Author):

The authors have extensively revised their manuscript and have done an excellent job of addressing the concerns raised during the review of their original manuscript. Thus, the manuscript is now acceptable for publication.

Reviewer #2 (Remarks to the Author):

The authors have revised the manuscript and successfully addressed some of the concerns raised by the reviewer, which significantly contributed to the coherence and clarity of the manuscript. It's also highly appreciated that further MS/MS experiments have been carried out for the proper identification of the peaks detected in the imaging experiment. However the major performance characteristics of the method have not changed, the method as presented does not compare favorably to SIMS or AP-MALDI with regard to spatial resolution or the biochemical information provided. To re-emphasize these problems, the mass spectrometric imaging resolution is defined by the size of the smallest biological feature detected. The additional experiments using a mesh have not revealed any convincing evidence for the 2-5 um spatial resolution claimed by the authors. The data still suggests tens of micrometers at its best, which is substantially worse than the 2-5 um convincingly demonstrated by Spengler et al. in a number of publications. Furthermore, both MALDI and SIMS can detect orders of magnitude more intact biochemical species at similar resolutions. This is not a battle of numbers, but the amount of biochemical information provided by the method. The described approach gives information on a few molecular fragments (according to Table 1 on 11 species) without knowing their exact origin, i.e. MAGs are general thermal degradation products of any glycerolipids. Nevertheless, the technique is interesting and certainly worth publishing in a journal aimed at the analytical chemistry community, e.g. Analytical Chemistry or the Analyst.

In conclusion I do not suggest the acceptance of the manuscript for publication.

Reviewer #3 (Remarks to the Author):

After a careful revision of the manuscript, I am satisfied with the modifications made by the authors and recommend publication.

Stéphane Larochelle, PhD
Senior Editor
Nature Communications

16th-October-2017

Nature Communications, *Manuscript Resubmission; NCOMMS-17-00257A-Z*

Dear Editor,

We are resubmitting our manuscript, entitled “*Atmospheric Pressure Mass Spectrometry Imaging of Live Hippocampal Tissue Slices with Subcellular Spatial Resolution*” for publication in *Nature Communications*. This manuscript is the revised version of our previous manuscript (*NCOMMS-17-00257A-Z*).

Below, we addressed each of the comments and suggestions raised by the reviewer. We are attaching the revised manuscript with the changes showing in blue.

Thank you again for reconsidering our manuscript.

Sincerely yours,

DaeWon Moon, Ph. D.

Professor, Department of New Biology,

Daegu Gyeongbuk Institute of Science & Technology (DGIST)

Dalseong, Daegu, 711-873, Republic of Korea

Phone: 82-10-5432-5390, E-mail: dwmoon@dgist.ac.kr

Answers to Reviewer's Comments

Title: “*Atmospheric Pressure Mass Spectrometry Imaging of Live Hippocampal Tissue Slices with Subcellular Spatial Resolution*”

Manuscript ID: NCOMMS-17-00257A-Z

We found the Reviewer's comments very helpful in refining the manuscript. Below are our point-by-point responses to the Reviewers' comments.

I. Reviewer # 2 comments

The authors have revised the manuscript and successfully addressed some of the concerns raised by the reviewer, which significantly contributed to the coherence and clarity of the manuscript. It's also highly appreciated that further MS/MS experiments have been carried out for the proper identification of the peaks detected in the imaging experiment.

- We appreciate your review and comments. We believe that we had considered and responded very sincerely to answering your comments.

However the major performance characteristics of the method have not changed, the method as presented does not compare favorably to SIMS or AP-MALDI with regard to spatial resolution or the biochemical information provided. To re-emphasize these problems, the mass spectrometric imaging resolution is defined by the size of the smallest biological feature detected. The additional experiments using a mesh have not revealed any convincing evidence for the 2-5 um spatial resolution claimed by the authors. The data still suggests tens of micrometers at its best, which is substantially worse than the 2-5 um convincingly demonstrated by Spengler et al. in a number of publications.

- Since a mouse hippocampus is a quite homogeneous tissue composed of several neuronal types, the tissue slice specimen has neither clear boundary nor distinct small biological feature. In addition, since the hippocampal tissue slice employed in this study was not co-crystallized by matrix and kept in humid during analysis, we cannot find proper cracks on the surface, which shows that this specimen is not suitable for the lateral resolution testing based on cracks formed during sample drying. The lack of cracks demonstrate that our method of live tissue imaging does not introduce any artifacts or distortions from the original image.

Thus, we analyzed a caudal fin of zebrafish (*Danio rerio*) as a bio-specimen to measure spatial resolution. The caudal fin consists of structurally separated bony rays,

blood vessels, and inter-ray mesenchymal tissue. When the mass spectrometry analysis is performed, the adenine peak ($m/z = 136.061$) has a very distinct boundary around the bone ray area, thus the lateral resolution can be measured with this boundary. As a result, the lateral resolution is measured to 2.1 – 3.1 μm when the x-axis direction velocity is 10 $\mu\text{m/s}$, which shows that the results are not significantly different from those when using the TEM grid with 400 mesh in **Fig. 5** of the manuscript. In fact, it is not unusual to use a spatially defined grid to test lateral spatial resolution in mass spectrometry imaging technique.^{R1,R2}

Therefore, without eliminating **Fig. 5** of the manuscript, the enlarged graphs of **Fig. 5c** are added in Supplementary Information (**Supplementary Fig. 7**) due to the length limit of the manuscript. The existing manuscript has already slightly exceeded the length limit given in the Nature Communications' guidelines. Thus, we added the results of lateral resolution obtained from the experiment of the caudal fin of the zebrafish to the Supplementary Information (**Supplementary Fig. 8**) and the number of Supplementary Figures is re-arranged.

- R1. Kiss, A. *et al.* Cluster SIMS microscope mode mass spectrometry imaging; supporting information part. *Rapid Commun. Mass Spectrom.* **27**, 2745–2750 (2013).
- R2. Cui, Y. *et al.* High lateral resolution vs molecular preservation in near-IR fs-laser desorption postionization mass spectrometry. *Anal. Chem.* **87**, 367–371 (2015).

Thus, the sentence at lines 8–10 on page 11 is changed in the new manuscript as follows:

“The x-axial lateral resolution of the MAG (16:0) ion image has been derived from the average of the three neighboring scan lines, as shown in **Fig. 5c,d** and **Supplementary Fig. 7.**”

The sentence at lines 15–17 on page 11 is added in the new manuscript as follows:

“We also confirmed similar micrometer lateral resolution from MS imaging of the caudal fin of zebrafish (see **Supplementary Fig. 8.**)”

Supplementary Fig. 7 The x-line scan for MAG (16:0) ion at $m/z = 313.273$ measured by the hippocampal tissue slice on a 400 mesh grid with the sample moving velocities of $10 \mu\text{m s}^{-1}$ (**Fig. 5c** in the manuscript) and enlarged graphs of (i) the rising slope and (ii) the falling slope.

Supplementary Fig. 8 The x-axial lateral resolution test of AP-nanoPALDI imaging in (a, b) a caudal fin of zebrafish (*Danio rerio*). Scale bar, 100 μm . (c) Ion image for the adenine ion at $m/z = 136.061$ was taken during the analysis of the caudal fin (the area of $600 \mu\text{m} \times 500 \mu\text{m}$) when the sample moving velocities were set to $10 \mu\text{m s}^{-1}$, respectively. (d, e) Enlarged graphs of the x-line scan of (i) the falling slope and (ii) the rising slope of the ion image.

Furthermore, both MALDI and SIMS can detect orders of magnitude more intact biochemical species at similar resolutions. This is not a battle of numbers, but the amount of biochemical information provided by the method. The described approach gives information on a few molecular fragments (according to Table 1 on 11 species) without knowing their exact origin, i.e. MAGs are general thermal degradation products of any glycerolipids. Nevertheless, the technique is interesting and certainly worth publishing in a journal aimed at the analytical chemistry community, e.g. Analytical Chemistry or the Analyst .

- I appreciate your helpful comments. As commented by the reviewer, detecting molecular ion peaks of biomolecules instead of molecular fragments will be more desirable method in the mass spectrometric analysis. I do not agree on this comment that our approach gives information on a few molecular fragments (according to Table 1 on 11 species). Most of them are molecular ions protonated or dehydrated. Additional experiments on DAG standard samples showed fragment ions and molecular ions with intensities of the same orders of magnitude. Lack of clear DAG molecular ions from hippocampal tissues suggest that most of the observed MAG ions are originated from MAG molecules. Identifying specific compositions of MAGs with spatial information is also give us valuable information.^{R3,R4}

For brain tissue imaging with mass spectrometry technique, it is important to observe that there are several types of lipids, such as glycerol lipids, sterol lipids, and sphingolipids as tabulated in **Table 1** in the neuronal cells differentiated into arbor structures and the apical and basal dendrites have a different distribution of these lipids (Ref. 54 in the manuscript). Because the lipid composition of apical and basal dendrites provides important information that tell how a sub-region of the hippocampus is chemically coordinated with other sub-regions as shown in **Fig. 4g** and the related description in the manuscript. This is clearly demonstrated with our AP-nanoPALDI system. We have already described the biological significance of the MS information we obtained using the proposed AP-MS method in detail in the Discussion section.

The most important point of our technique is live tissue imaging without sample treatment such as freezing, drying, fixation, and matrix application. I'd like to emphasize that all the biochemical information we obtained from this approach is not

distorted by sample treatment, which is the most critical contribution I can claim for novel mass spectrometric imaging methodology for innovative biology and medicine science.

We are also pursuing the improved method which possible to detect the molecular ion peaks having a higher m/z by changing the ionization method such as ESI, this will be the future direction of current research.

We greatly appreciate your insightful comment.

R3. Gao, F. *et al.* Monoacylglycerol analysis using MS/MS^{ALL} quadruple time of flight mass spectrometry. *Metabolites* **6**, 25 (2016).

R4. Kalo, P. J. *et al.* Identification of molecular species of simple lipids by normal phase liquid chromatography–positive electrospray tandem mass spectrometry, and application of developed methods in comprehensive analysis of low erucic acid rapeseed oil lipids. *Int. J. Mass Spectrom.* **254**, 106–121 (2006).

Answers to Editor's Suggestions

This manuscript has been transferred from *Nature Methods*. Therefore, the format was slightly different from the manuscript guideline of *Nature Communications* and was revised based on the manuscript checklist of *Nature Communications* according to the editor's request as follows:

1. According to the manuscript checklist, the abstract of the paper was reduced to 148 words and the main text (Introduction, Results, and Discussion) was reduced to 4994 words in the new manuscript.
2. We corrected the section order and eliminated the conclusion section.
3. The unit dimensions changed to negative integers in the new manuscript and the supplementary note.
4. In the discussion section, we eliminated the subheadings and clarified the content so that it does not overlap with the content of the results section.
5. The description about the preparation of the spatial resolution test specimen in the discussion section has been moved to the methods section.
6. The number of references has been reduced from 76 to 70. References 61, 69, 70, 71, 74, and 75 in the old manuscript were eliminated and the references have been renumbered.
7. The scale bar did not label within the figures (**Fig. 3**, **Fig. 4a,b**, **Fig. 5a,b**, **Fig. 6g**, and **Fig. 7a,b** in the manuscript and **Supplementary Fig. 4b**, **Fig. 8b**, **Fig. 9**, **Fig. 10a,b**, and **Fig. 11** in Supplementary Information) and was described in the legend.
8. The legend of **Table 1** was moved underneath the table.
9. Two supplementary figures were added (**Supplementary Fig. 7** and **8**) in the revision process by the reviewer's comment and the Supplementary Figures have been renumbered.

10. Our paper mainly is mainly about the method of analytical chemistry, many parts of “Reporting Checklist For Nature Communications Life Sciences Articles” do not apply to the manuscript. However, since the analytical specimen uses a tissue slice from a mouse, I have faithfully filled out the part related the manuscript (#1, #5, #8, and #9) in the life sciences checklist.

I attached “Nature Communications Manuscript Checklist” and “Reporting Checklist For Nature Communications Life Sciences Articles” as well.

I really appreciate your kindness and suggestion.

Reviewers' Comments:

Reviewer #2 (Remarks to the Author):

The authors revised the manuscript along the lines of the reviewer's requests. They provided further information on the lateral resolution of the method and the potential origin of the detected species. Regarding lateral feature resolution, the commonly accepted definition is the diameter of the smallest feature of biological origin. In that regard the data provided shows rather 10 um than 2 um resolution. Also, the reasoning on the origin of DAG and MAG species is not convincing as they should have used isotope labelled standards of other glycerolipids in tissue matrix. Nevertheless, the reviewer's main problem was that the authors present an approach which does not have better resolution than other AP MSI methods, but the biochemical information content is much lower. The authors fail to show any competitive advantage of the method over others and also fail to show any new biological information which can only be obtained using their novel approach. Again, the method is certainly interesting from a purely analytical point of view and should be published in an analytical chemistry or mass spectrometry journal.